
# Dark matter in anomaly-free gauge extensions

**Martin Bauer[1], Sascha Diefenbacher[1], Tilman Plehn[1]\*,
Michael Russell[1] and Daniel A. Camargo[2]**

**1** Institut für Theoretische Physik, Universität Heidelberg, Germany
**2** International Institute of Physics, Universidade Federal do Rio Grande do Norte,
Campus Universitario, Lagoa Nova, Natal, Brazil

\* plehn@uni-heidelberg.de

## Abstract

A consistent model for vector mediators to dark matter needs to be anomaly-free and include a scalar mode from mass generation. For the leading U(1) extensions we review the structure and constraints, including kinetic mixing at loop level. The thermal relic density suggests that the vector and scalar masses are similar. For the LHC we combine a $Z'$ shape analysis with mono-jets. For the latter, we find that a shape analysis offers significant improvement over existing cut-and-count approaches. Direct detection limits strongly constrain the kinetic mixing angle and we propose a $\ell^+\ell^-\not{E}_T$ search strategy based on the scalar mediator.

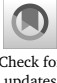

**Content**



# 1 Introduction

The nature of dark matter is one of the great mysteries in particle physics and cosmology. A comprehensive experimental program is on the way to identify the dark matter agent and determine its properties. On the theory side, many years of intense research have convinced us that perturbative gauge theories are the appropriate framework to describe physics above the QCD scale. The leading dark matter candidate around the weak scale is thermal freeze-out dark matter [1–6], naturally predicting the observed relic density for weak-scale or TeV-scale masses and electroweak-sized couplings.

The Standard Model allows for three renormalizable couplings to a mediator to the dark sector: The Higgs portal, the neutrino portal and the vector portal. The vector portal predicts a new spin-1 $Z'$ boson that couples to SM matter through a kinetic mixing term and can be searched for at colliders [8–16,16–25,27–29]. For order one gauge couplings, the approximate relation $\Omega_\chi h^2 \approx 5 \cdot 10^{-10}\,\text{GeV}^{-2}/\langle \sigma_{\chi\chi} v \rangle$ relates the observed relic density to a (large) dark matter annihilation rate. For example in the case of $m_{Z'} = m_\chi \dots 2m_\chi$ this turns into the condition

$$\langle \sigma_{\chi\chi} v \rangle \approx \frac{g^4 m_\chi^2}{16\pi m_{Z'}^4} \quad \Rightarrow \quad \frac{m_{Z'}}{g^2} < 1\,\text{TeV}\,, \tag{1}$$

with a dark matter mass $m_\chi$, a mediator mass $m_{Z'}$, and a perturbative coupling $g$. Similarly, for even heavier mediators with on-shell decays to the dark matter agent, $m_{Z'} > 2m_\chi$, we find

$$\langle \sigma_{\chi\chi} v \rangle \approx \frac{g^4}{16\pi m_{Z'}^2} \quad \Rightarrow \quad \frac{m_{Z'}}{g^2} < 2\,\text{TeV}\,. \tag{2}$$

This mediator mass range implies that a global analysis of thermally produced dark matter with the observed relic density is described by a fully propagating mediator at the LHC [31,32].

Besides a mediator that only communicates with Standard Model matter through kinetic mixing, SM particles could be gauged under the new gauge group. In this case a consistent ultraviolet complete model requires possible gauge anomalies to cancel. [33–37], for the generators of the gauge group and a sum or trace over the relevant left-handed fermions. If the $Z'$ mass is generated by a Higg-mechanism, the corresponding scalar can play an important role in phenomenology. Such a scalar is usually omitted in simplified models, in spite of the absence of any formally applicable decoupling argument [38]. Following both these arguments, an appropriate simplified model of a heavy spin-1 mediator includes

- a gauge boson and a scalar describing the massive mediator sector [39];
- either new fermions or an anomaly-free gauge group [40–47].

Both of these aspects need to be considered when we construct meaningful models for dark matter with vector mediators.

Anomaly-free gauge groups therefore provide well motivated mediators to dark matter. More general models are possible, but predict a sizable number of new fermions to cancel the anomalies, which contribute to interactions between the dark sector and the Standard Model [33–36]. We focus on anomaly-free gauge mediators based on the three possible setups:

1. We assign all SM fermions as singlets under the new group $U(1)_X$ and only charge the dark matter fermion, which in turn does not couple to the SM gauge bosons. This setup is trivially free of anomalies, and the $Z'$ couplings to Standard Model fermions arise through kinetic mixing [48–57].

2. We choose an anomaly-free gauge group based on lepton number and utilize more than one generation for the anomaly condition. Viable examples are the charged lepton number differences $U(1)_{L_\tau - L_\mu}, U(1)_{L_\tau - L_e}$, or $U(1)_{L_\mu - L_e}$ [58–68]. Such models can be motivated for example through neutrino masses [69–75] or flavor anomalies [76–80]. The corresponding baryon-number-based constructions are usually ruled out by the observed structure of the CKM matrix.

3. We gauge the difference between baryon and lepton number $U(1)_{B-L}$ [81–86]. It has the specific advantage of allowing for Majorana masses for right-handed neutrinos after symmetry breaking [87]. In that sense, an anomaly-free $U(1)_{B-L}$ gauge group is motivated by a structural deficit of the Standard Model, because it requires right-handed leptons at some scale.

We argue that searches for missing energy signals at the LHC are particularly powerful for two of these models, namely the $U(1)_X$ and the $U(1)_{L_\mu - L_\tau}$ gauge groups. After deriving the properties of the mediators for the three classes of models defined above, we focus our analysis on these two models. Other known anomaly-free $U(1)$ extensions include c $U(1)_{L_\mu + L_\tau - 2L_e}$ or $U(1)_R$, where right-handed SM fields carry charges proportional to $T^3$ of $SU(2)_R$. However, their phenomenology is not expected to be fundamentally different from the three above cases, and in some cases the structure is actually equivalent [88].

In this paper we will first introduce a kinetically mixed gauge extension and the Higgs-like scalar in Sec. 2 and discuss the three anomaly-free gauge extensions we focus on in the remainder of this paper. This includes not only the general structure of the model, but also the decay modes of the heavy gauge bosons and their Higgs-like scalars. In Sec. 3 we will collect all available constraints from low-energy and collider data. The properties of the new particle as a dark matter mediator will be the focus of Sec. 4. Finally, we will compare different LHC strategies for searching for the new heavy states in Sec. 5.

## 2  U(1)-gauge extensions

We consider consistent dark matter models with a spin-1 mediator $Z'$ and a dark matter fermion $\chi$, charged under the new gauge group. The available options are purely singlet SM fermions, gauged lepton number differences, or the well-known anomaly-free difference

between the lepton and baryon numbers [89],

$$U(1)_X\,, \qquad\qquad U(1)_{L_i-L_j}\,, \qquad\qquad U(1)_{B-L}\,, \qquad\qquad (3)$$

with $i \neq j = 1,2,3$. The $Z'$ couplings to currents of SM fermions are given by

$$
\begin{aligned}
\mathcal{L}_{\text{fermion}} &= -g_{Z'} j'_\mu Z'^\mu \\
j'_\mu &= 0 & U(1)_X \\
j'_\mu &= \bar{L}_i \gamma_\mu L_i + \bar{\ell}_i \gamma_\mu \ell_i - \bar{L}_j \gamma_\mu L_j - \bar{\ell}_j \gamma_\mu \ell_j & U(1)_{L_i-L_j} \\
j'_\mu &= \frac{1}{3}\bar{Q}\gamma_\mu Q + \frac{1}{3}\bar{u}_R \gamma_\mu u_R + \frac{1}{3}\bar{d}_R \gamma_\mu d_R - \bar{L}\gamma_\mu L + \bar{\ell}\gamma_\mu \ell & U(1)_{B-L}\,, \quad (4)
\end{aligned}
$$

where $g_{Z'}$ denotes the dark gauge coupling. The different coupling structures shown above can be understood in terms of a flavor structure of a dark gauge coupling matrix.

The fermion current structure of Eq.(4) can be generalized to include the dark matter current. To couple to the gauge mediator the dark matter fermion has to be a Dirac fermion. To avoid new anomalies, the dark matter candidate cannot be chiral and its charges under the new gauge group are $q_{\chi_L} = q_{\chi_R}$. This defines a dark fermion Lagrangian with a vector mass term

$$\mathcal{L}_{\text{DM}} = i\bar{\chi}\slashed{D}\chi - m_\chi \bar{\chi}\chi\,, \qquad\qquad (5)$$

with the covariant derivative of the SM-singlet fermion $D_\mu = \partial_\mu - ig_{Z'}q_\chi \hat{Z}'_\mu$.

In all cases, the kinetic term for the $U(1)$ gauge bosons is not canonically normalized

$$\mathcal{L}_{\text{gauge}} = -\frac{1}{4}\begin{pmatrix} \hat{B}_{\mu\nu} & \hat{Z}'_{\mu\nu} \end{pmatrix}\begin{pmatrix} 1 & s_{Z'} \\ s_{Z'} & 1 \end{pmatrix}\begin{pmatrix} \hat{B}_{\mu\nu} \\ \hat{Z}'_{\mu\nu} \end{pmatrix}\,, \qquad\qquad (6)$$

and afternormalizing the kinetic terms and rotating to the mass eigenbasis, the masses of the vector bosons are given by

$$m_\gamma = 0$$

$$m_Z^2 = \frac{v^2}{4}(g^2 + g'^2)\left(1 - \frac{v^2}{v_S^2}\frac{s_{Z'}^2 g'^2}{8g_{Z'}^2 q_S^2}\right) + \mathcal{O}\left(\frac{v^6}{v_S^4}\right) \qquad\qquad (7)$$

$$m_{Z'}^2 = \frac{g_{Z'}^2 q_S^2 v_S^2}{2c_{Z'}^2} + \frac{v^2}{4}g'^2 t_{Z'}^2 + \mathcal{O}\left(\frac{v^4}{v_S^2}\right)\,. \qquad\qquad (8)$$

For details of the calculation, we refer the reader to Appendix A.

As a second structural ingredient we give mass to the new gauge boson by introducing a complex scalar $S$ with the potential

$$\mathcal{L}_{\text{scalar}} = \frac{1}{2}(D_\mu S)(D^\mu S)^\dagger + \mu_S^2 S^\dagger S + \frac{\lambda_S}{2}(S^\dagger S)^2 + \lambda_{HS} H^\dagger H S^\dagger S\,. \qquad (9)$$

In this case the covariant derivative introduces the charge $q_S$ of the heavy scalar under the new gauge group.

Under the conservative assumption that SM gauge couplings and the Higgs vacuum expectation value are fixed, the error $m_Z = 91.1876 \pm 0.0021$ [90] constrains the mixing to

$$\frac{g_{Z'}q_S}{s_{Z'}}v_S \gtrsim 1.3\,\text{TeV} \qquad \text{at} \quad 95\%\,\text{CL}\,. \qquad\qquad (10)$$

It is interesting to compare the mass parameters for the heavy new scalar and the heavy new vector modes in the mass matrices of Eq.(43) and Eq.(47)

$$\frac{m_S}{m_{Z'}} \sim \frac{\sqrt{\lambda_S}}{g_{Z'}q_S/c_{Z'}} , \tag{11}$$

where we identify the heavy entries in the mass matrices with the new masses and ignore parameters which are expected to be of order one. Separating these two mass scales is not impossible, but requires a dedicated model building effort, which means that a generic analysis of gauge extensions should include the scalar mode in the mediator sector.

The couplings of the mass eigenstates to fermions and scalars play an important role in the following analysis and we find

$$\begin{aligned}
\mathcal{L}_{\text{fermion}} = {} & e j_{\text{em}} A \\
& - c_w s_3 t_{Z'} e j_{\text{em}} Z + (c_3 + s_w s_3 t_{Z'}) \frac{e}{s_w c_w} j_Z Z + \frac{s_3}{c_{Z'}} g_{Z'} j_{Z'} Z \\
& - c_w c_3 t_{Z'} e j_{\text{em}} Z' + (s_w c_3 t_{Z'} - s_3) \frac{e}{s_w c_w} j_Z Z' + \frac{c_3}{c_{Z'}} g_{Z'} j_{Z'} Z'
\end{aligned} \tag{12}$$

and

$$\begin{aligned}
\mathcal{L}_{\text{scalar}} \ni {} & \frac{v}{8} (g^2 + g'^2)(c_\alpha H - s_\alpha S) Z_\mu Z^\mu \\
& + \frac{v}{4} s_w t_{Z'} (g^2 + g'^2)(c_\alpha H - s_\alpha S) Z_\mu Z'^\mu \\
& + \frac{v}{8} s_w^2 t_{Z'}^2 \left[ c_\alpha \left( g^2 + g'^2 + \frac{4 g_{Z'}^2 q_S^2 t_\alpha}{s_w^2 s_{Z'}^2} \frac{v_S}{v} \right) H - s_\alpha \left( g^2 + g'^2 - \frac{4 g_{Z'}^2 q_S^2 t_\alpha}{s_w^2 s_{Z'}^2} \frac{v_S}{v} \right) S \right] Z'_\mu Z'^\mu .
\end{aligned} \tag{13}$$

The phenomenology of anomaly-free $U(1)$-extensions can thus be described by a small number of model parameters. The Lagrangian features the most relevant new parameters

$$\{ m_\chi, g_{Z'}, m_{Z'}, s_{Z'}, m_S, \lambda_{HS} \} . \tag{14}$$

The charges under the new $U(1)$-symmetry we assume to be of order one. As long as we focus on a heavy dark matter mediator with on-shell decays, $m_{Z'} > 2 m_\chi$, the dark matter mass mainly enters the computation of the mediator widths $\Gamma_{S,Z'}$.

The vector and scalar mediator masses are typically related, as shown in Eq.(11). A hierarchy with a comparably light scalar $\lambda_S \ll g_{Z'}$ is possible, but not the focus of our paper. Alternatively, the scalar can be heavier than the vector, $g_{Z'} \ll \lambda_S < 4\pi$. In this case, the small gauge coupling suppresses the interaction of the new gauge boson with the Standard Model. This does not only affect the LHC production cross section, it also reduces the annihilation cross section in the early universe to the point where an efficient annihilation is only possible around the pole condition $m_{Z'} = 2 m_\chi$.

The phenomenology of the vector mediator is determined by its couplings to the Standard Model and by its mass $m_{Z'}$. In Eq.(50) we see that couplings to SM fermions can arise through kinetic mixing ($t_{Z'}$), through mixing with the $Z$-boson ($s_3$), or through the $U(1)$ charges of the fermions ($g_{Z'}$).

The properties of the new scalar $S$ are largely independent of the dark matter properties. All couplings to a pair of SM particles proceed through the Higgs portal ($s_\alpha$), with the possible exception of a the coupling to right-handed neutrinos in the case of $U(1)_{B-L}$. Interesting features only arise in couplings linking both mediators, like the $Z'$-$S$-$Z$ coupling.

## 2.1 $U(1)_X$

In our first setup all SM particles are singlets under the new $U(1)$ gauge symmetry and all $Z'$ and $S$ couplings to the Standard Model are induced by mixing. Both of these mixing effects lead to a transition between the SM-sector and the dark matter sector of the theory. The relation between the kinetic mixing in the gauge sector and the scalar mixing in the Higgs sector reflects a symmetry structure reminiscent of gaugino and scalar masses in broken supersymmetry [93–95]. While the gauge-kinetic mixing is protected by the additional gauge coupling and only multiplicatively renormalized, the scalar mixing is just a property of the Higgs potential. This is why quantum effects transform gauge-kinetic mixing into a finite scalar mixing, but scalar mixing does not induce mixing in the gauge sector.

As a starting point, we show the $Z'$ branching ratios in Fig. 1, assuming two values of sizeable kinetic mixing. Clearly, $Z'$ decays to two dark matter fermions through an un-suppressed $U(1)_X$ charge dominates, provided the process $Z' \to \chi\bar{\chi}$ is kinematically allowed. The partial widths to SM fermions are universally proportional to $s_{Z'}$, including the $Z' \to \nu\bar{\nu}$ background to the dark matter signal. Due to the non-orthogonal mixing in Eq.(50) the electromagnetic current contributes, so the structure of the $Z'$ branching ratios does not correspond to $Z$-decay channels. Decays to light quark pairs reach branching ratios of 30% ... 40%, enhanced by color factors. Decays to leptons amount to almost the same rate. The $t\bar{t}$ decay channel exceeds 10% slightly above its threshold.

The bosonic decays $Z' \to SZ, HZ$ can reach per-cent-level branching ratios. Other bosonic channels, like $Z' \to ZZ$ or $Z' \to HH$ are not possible. For the two dominating bosonic channels we find that the leading diagrams lead to a scaling

$$\frac{\text{BR}(Z' \to SZ)}{\text{BR}(Z' \to HZ)} \propto t_\alpha^2, \tag{15}$$

with the larger $\text{BR}(Z' \to ZH)$ at the per-cent level. The mixing angle $s_\alpha$ changes for the different values of $s_{Z'}$ used in Figs. 1 and 3, because $v_S$ depends on this choice when all other parameters remain the same. While one might expect effects from mass insertions $1/m_Z$ or $1/m_{Z'}$ in this ratio, the corresponding diagrams for the decay $Z' \to SZ$ are sub-leading for

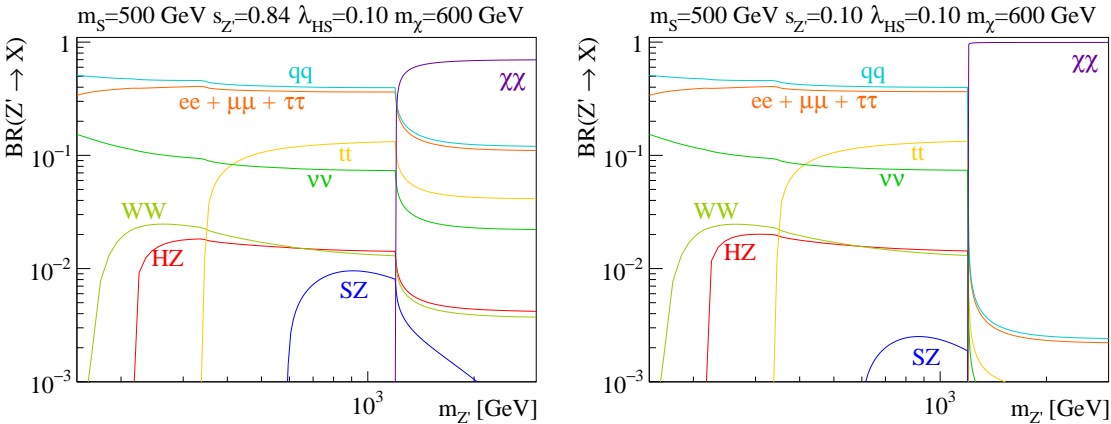

Figure 1: Branching ratios for the $U(1)_X$ gauge boson with $m_S = 500$ GeV, $\lambda_{HS} = 0.1$, $g_{Z'} = 1$, and $s_{Z'} = 0.84$ (left) and $s_{Z'} = 0.1$ (right). The variable $m_{Z'}$ is varied through $v_S = 50 - 1150$ GeV in the left panel and $v_S = 100 - 2110$ GeV in the right panel. Correspondingly, the Higgs mixing angle varies between $s_\alpha = 0.001 - 0.12$ (left) and $s_\alpha = 0.008 - 0.23$.

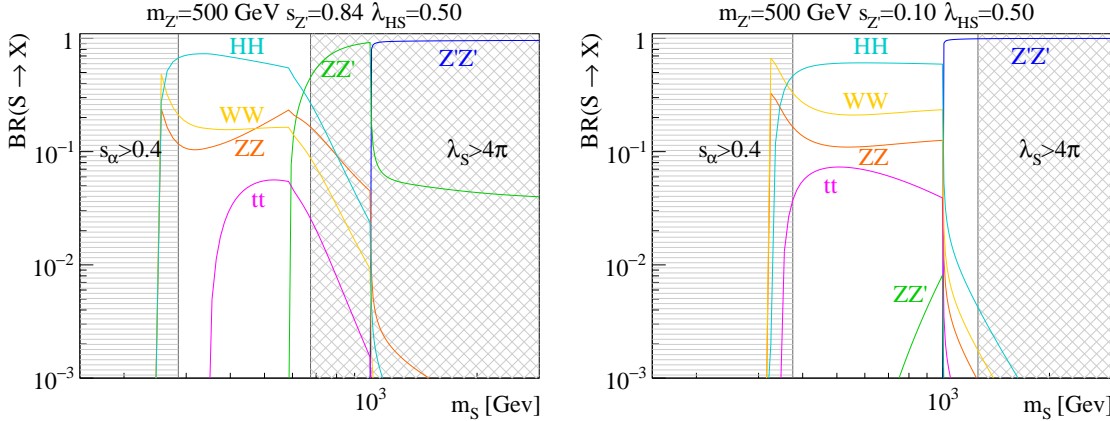

Figure 2: Branching ratios for the heavy scalar $S$ with $m_{Z'} = 500$ GeV, $\chi = 1$ and $g_{Z'} = 1$ and for different values of the quartic coupling $\lambda_{HS}$. The hashed regions are excluded by Higgs signal strengths measurements ($s_\alpha > 0.4$) and perturbativity of the scalar potential ($\lambda_S > 4\pi$).

finite Higgs mixing. As we will see, for sizable kinetic mixing the decay $Z' \to HZ$ combined with a large invisible decay rate provides a tell-tale signal of this type of models.

For small kinetic mixing, the only way to produce the $Z'$ with a sizeable rate is $S$-production with a decay $S \to Z'Z'$. The branching ratios of the heavy scalar $S$ for small and large gauge mixing $s_{Z'}$ are shown in Fig. 2. We indicate constraints by perturbativity, $\lambda_S > 4\pi$, and global Higgs analysis results, $s_\alpha < 0.4$ at 68% C.L. [91, 92]. Decays to the SM Higgs boson or SM gauge bosons, mediated by the Higgs portal, dominate over a wide range of parameters. The mixed decay $S \to ZZ'$ turns on for large kinetic mixing $s_{Z'}$, but for both choices of $s_{Z'}$ the direct decay to $Z'Z'$ pairs completely dominates once it is allowed. The only caveat is that in this regime the self-coupling $\lambda_S$, responsible for the mass of the heavy scalar, can become very large.

Finally, looking at the models there exists a fundamental difference between the kinetic gauge mixing and the Higgs mixing. If the $U(1)_X$ group is embedded in a non-abelian gauge group $SU(N)_X$ at a higher scale, kinetic mixing is never generated. On the other hand no symmetry principle forbids a Higgs portal. In this limit, our $U(1)_X$ model corresponds to a Higgs-portal model with an dark sector consisting of the vector $Z'$ and the fermions $\chi$. The main signature is $pp \to S$ production with an invisible decay to $Z'Z' \to 4\chi$.

In essence, we find that the $Z'$ typically decays to SM fermions, including a large branching ratio to leptons. If kinematically possible, the invisible decay to dark matter will dominate, especially for small mixing $s_{Z'} \lesssim 0.1$. In contrast, the new scalar prefers decays to SM Higgs and gauge bosons, unless the decay $S \to Z'Z'$ is kinematically allowed. The reason for this structure is that all $Z'$ couplings with the exception to dark matter are mediated by the mixing angle $s_{Z'}$. We will see that this structure inherently limits discovery prospects for this kind of dark matter mediator at colliders.

## 2.2 $U(1)_{L_\mu - L_\tau}$

Gauged differences of charged lepton numbers, such as $U(1)_{L_\mu - L_\tau}$, induce $Z'$ gauge couplings to charged and neutral leptons even for $s_{Z'} \to 0$. In return, SM lepton loops generate kinetic

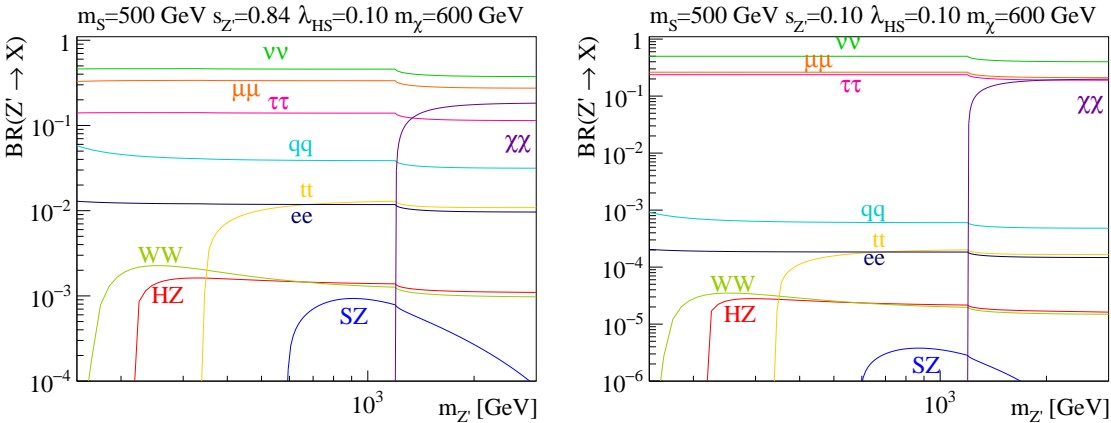

Figure 3: Branching ratios for the $U(1)_{L_\mu - L_\tau}$ gauge boson with $m_S = 500$ GeV, $\lambda_{HS} = 0.1$, $g_{Z'} = 1$, and $s_{Z'} = 0.84$ (left) and $s_{Z'} = 0.1$ (right).

mixing,

$$
\begin{aligned}
s_{Z'} &= -\frac{3g'g_{Z'}}{4\pi^2} \int_0^1 dx \; x(x-1) \log \frac{m_\tau^2 + q^2 x(x-1)}{m_\mu^2 + q^2 x(x-1)} \\
&= \begin{cases} \dfrac{g'g_{Z'}}{8\pi^2} \left( \dfrac{m_\tau^2}{q^2} - \dfrac{m_\mu^2}{q^2} \right) + \mathcal{O}\left( \dfrac{m_\tau^4}{q^4} \right) & \text{for } q^2 \gg m_\tau^2 \\[2ex] \dfrac{g'g_{Z'}}{8\pi^2} \log \dfrac{m_\tau^2}{m_\mu^2} + \mathcal{O}\left( \dfrac{q^2}{m_\mu^2} \right) \approx 0.025 \, g_{Z'} & \text{for } q^2 \ll m_\mu^2 \, . \end{cases}
\end{aligned}
\tag{16}
$$

Its size strongly depends on the energy scale at which we probe the $Z$-$Z'$ mixing. At large momentum transfer, like at the LHC, the mixing is dominated by the small ratio $m_\tau^2/q^2$. At low-energy experiments, like direct dark matter detection, both leptons can be integrated out and the remaining suppression is proportional to $\log m_\tau^2/m_\mu^2$. For an anomalous gauge group, this low-energy limit would not be defined and instead require an additional physical scale in the integral at which the anomaly is removed.

The fact that the loop-induced mixing is finite suggests that the $U(1)_{L_\mu - L_\tau}$ gauge group can be embedded into a gauge group which forms a direct product of $SU(3)_C \times SU(2)_L \times SU(N)$ [69–75]. While in the unbroken phase of the non-abelian group the additional condition $m_\mu = m_\tau$ removes this contribution, it appears in the broken phase with $U(1)_{L_\mu - L_\tau}$ intact.

In the absence of kinetic mixing all couplings are fixed by the charge assigned to the dark matter candidate. However, the LHC production rate scales like

$$
\sigma(pp \to Z') \propto s_{Z'}^2 \, .
\tag{17}
$$

Hence, once the model predicts a sizeable LHC rate, searches for a di-lepton resonance or for missing transverse energy are motivated by the leading $Z'$ branching ratios. They are shown in Fig. 3. For large mixing, the decays to muons and taus and their neutrinos dominate, but branching ratios to di-jets occur at per-cent level. Bosonic decays like $Z' \to HZ$ are rare, but will be useful to disentangle the origin of the $U(1)$ structure. The decay to two dark matter fermions opens above the kinematic threshold, but remains below the neutrino contribution to the combined invisible branching ratio. Reducing the mixing rapidly decouples all decay signatures, with the exception of $Z' \to \mu\mu, \tau\tau, \nu\nu$ and $Z' \to \chi\bar{\chi}$.

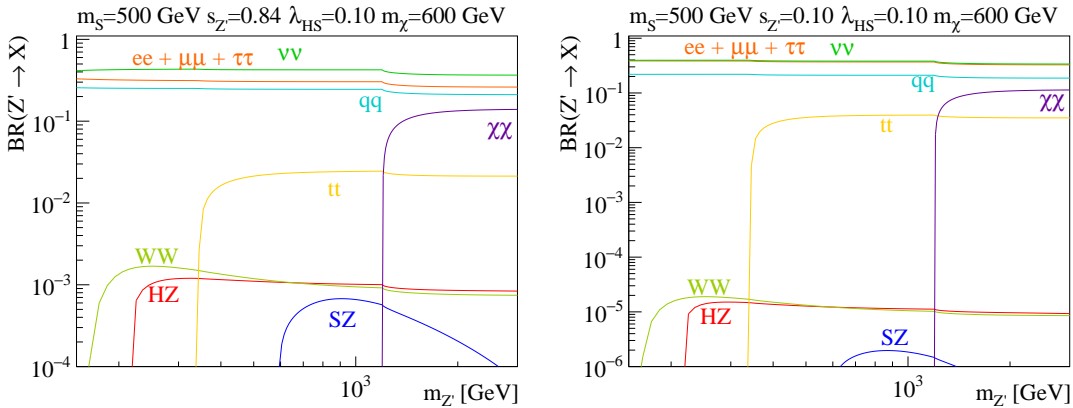

Figure 4: Branching ratios for the $U(1)_{B-L}$ gauge boson with $m_S = 500$ GeV, $\lambda_{HS} = 0.1$, $g_{Z'} = 1$, and $s_{Z'} = 0.84$ (left) and $s_{Z'} = 0.1$ (right).

Altogether, similar to the $U(1)_X$ case we find that di-lepton searches are the most promising ways to search for the $Z'$ boson. The difference to the $U(1)_X$ case is the absence of lepton universality, especially when it comes to electron couplings. In addition, the decay to dark matter only dominates over the mixing-induced decay channels, which also implies that the total invisible decay width tends to be dominated by $Z' \to \nu\bar{\nu}$.

## 2.3 $U(1)_{B-L}$

The $U(1)_{B-L}$ gauge symmetry predicts new gauge couplings to both quarks and leptons. Even for sizable kinetic mixing, the $Z'$ branching ratios shown in Fig. 3 are largely dictated by the charges,

$$\mathrm{BR}(Z' \to \ell^+\ell^-) : \mathrm{BR}(Z' \to q\bar{q}) : \mathrm{BR}(Z' \to \chi\bar{\chi}) \approx n_\ell : \frac{n_q N_c}{9} : q_\chi^2 , \tag{18}$$

where $N_c$ is a color factor and the factor $1/9$ accounts for the quark charges. We illustrate this scaling in Fig. 4. Searches for di-lepton and di-jet resonances are again promising. Even though a kinetic mixing of the kind shown in Eq.(16) is induced, such a contribution hardly changes the LHC search strategies, because tree-level generically beats loops. The only difference between the two panels in Fig. 4 is that the bosonic channels decrease from the per-mille level for large mixing to the $10^{-5}$ level for small mixing. Invisible decays of the heavy vector boson will typically also be dominated by $Z' \to \nu\nu$ decays, rather than decays to dark matter, $Z' \to \chi\bar{\chi}$.

From an LHC or relic density point of view the $U(1)_{B-L}$ scenario is attractive, because the $Z'$ mediator has sizeable gauge couplings to all fermions. For the phenomenology the universal mixing contribution $s_{Z'}$ is generally sub-leading. The problem with this model is that according to Fig. 4 invisible $Z'$ decays are dominated by decays to neutrinos. This means that an discovery of a $Z'$ decaying invisibly might have nothing to do with dark matter.

# 3 Collider and low-energy constraints

New gauge bosons have motivated new physics searches for many decades. For our three anomaly-free $U(1)$-extensions we consider three different types of constraints: firstly, couplings to electrons are constrained by LEP searches for new gauge bosons; secondly, couplings

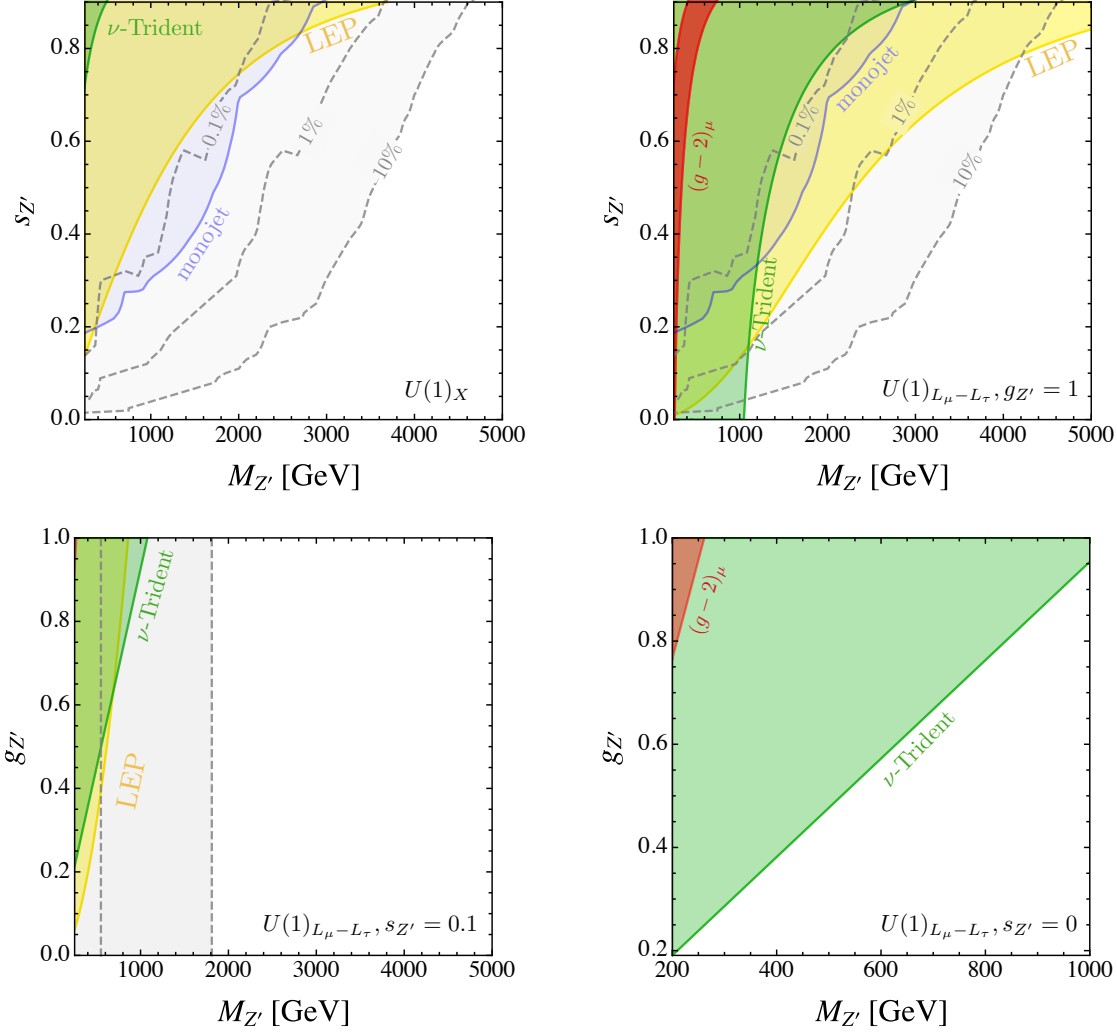

Figure 5: Bounds on the kinetic mixing angle $s_{Z'}$, the mass $m_{Z'}$, and the gauge coupling $g_{Z'}$ for a $U(1)_X$ gauge boson (upper left) and a $U(1)_{L_\mu - L_\tau}$ gauge boson with $g_{Z'} = 1$ (upper right) or $s_{Z'} = 0.1$ (lower left) and $s_{Z'} = 0$ (lower right).

to neutrinos lead to contributions to neutrino-nucleus scattering; finally, couplings to light-flavor quarks predict sizeable $Z'$ production rates at the LHC.

## 3.1 LEP

Obviously, LEP strongly constrains the couplings of a new gauge boson for $m_{Z'} \lesssim 209$ GeV. The luminosity at high energies translates into a limit on the kinetic mixing angle mediating the $Z'e^+e^-$ interaction, namely $s_{Z'} < 0.03$. This bound becomes stronger for a lighter $Z'$ [96].

Effects from heavier $Z'$ bosons can be described by effective 4-fermion interactions [97]

$$
\begin{aligned}
\mathcal{L}_{\text{eff}} = &- \frac{\kappa_H}{2\Lambda^2} |H^\dagger D_\mu H|^2 \\
&- \sum_{f,f'} \frac{4\pi \kappa_{ff'}}{\Lambda^2} (\bar{f}\gamma^\mu f)(\bar{f}'\gamma_\mu f') - \sum_f \left( \frac{i\kappa_{Hf}}{\Lambda^2} (\bar{f}\gamma^\mu f)(H^\dagger D_\mu H) + \text{h.c.} \right).
\end{aligned}
\tag{19}
$$

Any $Z'$ couplings involving the Higgs are either proportional to the scalar mixing angle $s_\alpha$ or of higher order in $v/v_S$ or $s_{Z'}$. As for searches for contact interactions, the strongest constraints

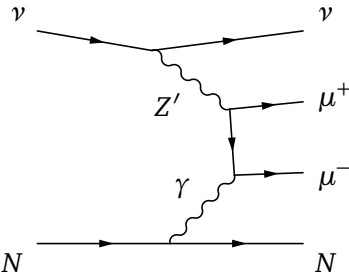

Figure 6: Example Feynman diagrams for neutrino trident production.

arise from $e^+e^- \to q\bar{q}, \ell^+\ell^-$ searches [98,99]. At 95% C.L. the LEP limits are

$$\frac{\Lambda}{\sqrt{\kappa_{\ell\ell}}} \gtrsim 24.5 \text{ TeV} \qquad\qquad \frac{\Lambda}{\sqrt{\kappa_{e\mu}}} \gtrsim 18.6 \text{ TeV}$$

$$\frac{\Lambda}{\sqrt{\kappa_{e\tau}}} \gtrsim 15.6 \text{ TeV} \qquad\qquad \frac{\Lambda}{\sqrt{\kappa_{eu}}} \gtrsim 14 \text{ TeV} . \qquad (20)$$

Matching to the full theory we identify the new physics scale as $\Lambda = \sqrt{8\pi} m_{Z'}$ and the Wilson coefficients $\kappa_{ff'}$ as functions of the $Z'$ couplings, the mixing angle $s_{Z'}$ and the chirality of the involved fermions. These LEP constraints put strong bounds on the new gauge bosons couplings to electrons [100, 101],

$$\frac{m_{Z'}}{g_{Z'}} > 6.9 \text{ TeV} \qquad\qquad\qquad U(1)_{B-L}$$

$$\frac{m_{Z'}}{g_{Z'}} > 5.25 \text{ TeV} \qquad\qquad\qquad U_{L_e-L_\mu}, U_{L_e-L_\tau} . \qquad (21)$$

These limits for $s_{Z'} = 0$ become even stronger for $s_{Z'} > 0$. The remaining parameter space will typically not give the observed relic density and push the additional scalar $S$ to large masses. This is why at this stage we will drop the $U(1)_{B-L}$ gauge group (and any other group with gauged electrons) from our analysis.

For the remaining gauge groups $U(1)_X$ and $U_{L_\mu-L_\tau}$ the coupling to leptons and with it the sensitivity to LEP constraints depends on the mixing angle. In Fig. 5 we show the excluded parameter space for $U(1)_X$ and $U_{L_\mu-L_\tau}$, the latter for fixed $g_{Z'} = 1$ or $s_{Z'} = 0.1$. First, we see that the the four-fermion constraints constrain both, the $U(1)_X$ and the $U_{L_\mu-L_\tau}$ models, unless $s'_Z = 0$. In terms of $m_{Z'}$ and $s_{Z'}$ the limits on both gauge groups are similar, because in both cases the electron couplings enters with $s_{Z'}$, but the muon coupling is a gauge coupling for $U_{L_\mu-L_\tau}$.

In the $U(1)_X$ model, the LEP constraints from contact interactions on $m_{Z'}$ and $s_{Z'}$ are similar in strength to the bound from the modification of the $Z$ mass, Eq.(10) [102], but stronger for the $U(1)_{L_\mu-L_\tau}$ gauge boson with a coupling $g_{Z'} \gtrsim 0.5$. Additional bounds arise from non-universal $Z$-couplings to muons and electrons This constraint is weaker than both the bounds from four-fermion interactions and from the $Z$-mass measurement, but in $U(1)_{L_\mu-L_\tau}$ a contribution arises at the one-loop level from $Z'$ exchange between the muon legs, which is present in the limit $s_{Z'} \to 0$ as well. The corresponding constraints are however weaker than the dominant constraint from neutrino-trident production discussed in the following section [103,104].

## 3.2 Low-energy probes

Additional gauge bosons are constrained by wealth of low-energy experiments. In our case, the $U(1)_{L_\tau-L_\mu}$ and $U(1)_{B-L}$ gauge bosons contribute to the production of $\mu^-\mu^+$ pairs in neutrino–

nucleus scattering or neutrino trident production

$$\nu_\mu N \to \nu_\mu N \, \mu^+ \mu^- \, , \tag{22}$$

shown in Fig. 6. The enhancement over the SM prediction for the total trident cross section to the SM prediction in the limit $m_{Z'} \gg m_\mu$, is [103, 104]

$$\frac{\sigma}{\sigma_{\rm SM}} = \frac{(1 + 2\,C_A^{Z'})^2 + (1 + 4 s_w^2 + 2\,C_V^{Z'})^2}{1 + (1 + 4 s_w^2)^2} \, , \tag{23}$$

with

$$C_V^{Z'} = \frac{v^2}{4 c_{Z'}^2 m_{Z'}^2}\left( 4 g_{Z'}^2 + 5 g' g_{Z'} s_{Z'} + \frac{3}{2} g'^2 s_{Z'}^2 \right), \tag{24}$$

$$C_A^{Z'} = \frac{v^2}{4 c_{Z'}^2 m_{Z'}^2}\left( g' g_{Z'} s_{Z'} + \frac{1}{2} g'^2 s_{Z'}^2 \right). \tag{25}$$

In our evaluation we neglect corrections $v/v_S$, and for the $U(1)_X$ gauge group all terms proportional to the new gauge coupling $g_{Z'}$ vanish. The combined measurement from CHARM-II [105] and CCFR [106] comes to

$$\frac{\sigma}{\sigma_{\rm SM}} = 0.83 \pm 0.28 \, . \tag{26}$$

We show the excluded parameter space in Fig. 5. For $U(1)_X$ the trident constraints are mediated by the two mixing vertices, so the cross section is suppressed by $s_{Z'}^4 \ll 1$. For $U(1)_{L_\mu - L_\tau}$ all vertices are new gauge couplings, so the trident constraint becomes much stronger and survives the limit $s_{Z'} \to 0$.

Interestingly, the $U(1)_{L_\mu - L_\tau}$ gauge boson can also provide an explanation of the long-standing discrepancy between the experimental value and the SM prediction for the anomalous magnetic moment of the muon [107–109]

$$a_\mu^{\rm exp} - a_\mu^{\rm SM} = (29.3 \pm 7.6) \times 10^{-10} \, . \tag{27}$$

The $Z'$ contribution in the limit $m_{Z'} \gg m_\mu$ is given by [110]

$$\Delta a_\mu = \frac{1}{48\pi^2} \frac{m_\mu^2}{m_{Z'}^2} \frac{1}{c_{Z'}^2}\left( g'^2 s_{Z'}^2 + 6 g' g_{Z'} s_{Z'} + 4 g_{Z'}^2 \right). \tag{28}$$

We show the preferred region shaded in red in Fig. 5. In all cases, this explanation is excluded for the masses we consider. Other low-energy constraints such as lepton flavor universality in $\tau$ decays or atomic parity violation do not yield additional constraints for the models and the parameter spaces we consider.

## 3.3 LHC resonance searches

Especially for heavier resonances, LHC searches for di-jet and di-lepton resonances constraint the mass range $m_{Z'} = 250 \dots 5000$ GeV [111, 113], provided there is a large enough coupling to the incoming quarks. In the case of the $U(1)_{B-L}$ gauge boson, the production cross section and all decay channels are sensitive to the universal coupling $g_{Z'}$ [114], unless $g_{Z'} \ll 1$ makes

it hard to obtain the correct dark matter abundance. For $U(1)_X$ or $U(1)_{L_\mu-L_\tau}$ gauge bosons the production cross section at the LHC depends on the kinetic mixing angle,

$$\sigma(q\bar{q} \to Z') = \frac{\pi^2}{12s} \frac{\alpha_e}{c_w^2} t_{Z'}^2 \sum_q \left(Q_q^2 + (T_3 - Q_q)^2\right) , \qquad (29)$$

where $Q_q$ and $T_3$ are the electric charges and the weak isospin of the quarks we neglect corrections of order $v^2/v_S^2$.

In Fig. 5 we include some approximate LHC limits for illustration. We compute the $Z'$ production cross section with MADGRAPH5 [115], accounting for higher order corrections using MATRIX [116, 117], estimating the NNLO effects by using the K-factor for the $Z$ boson Drell-Yan production cross section, and compare with the ATLAS di-lepton limits [111]. We take the branching ratio BR($Z' \to \ell^+\ell^-$) to be a free parameter and we show the excluded parameter space for values of 0.01% and 1% for $U(1)_X$ and 1% and 10% for $U(1)_{L_\mu-L_\tau}$. Especially in the $U(1)_{L_\mu-L_\tau}$ case a strong suppression of the decay $Z' \to \mu^+\mu^-$ rate can only be achieved through a large $U(1)_{L_\mu-L_\tau}$ charge of the dark matter candidate, leading to a Landau pole of the $U(1)_{L_\mu-L_\tau}$ gauge couplings at low energies.

In Fig. 5 we see that for small mixing angles all LHC constraints vanish, because the quarks are neutral under the two gauge groups. In the right panel we show that for a fixed, but small mixing angle the LHC production rate is fixed as well, and for a fixed branching ratio to leptons the allowed $Z'$ masses are typically in the TeV-range. Lighter new gauge bosons are only allowed for small mixing angles $s_{Z'} < 0.1$.

LHC searches for invisible $Z'$ decays will be discussed in Sec. 5.2. Searches for $Z \to 4\mu$ decays at the LHC can lead to additional constraints in the case of the $U(1)_{L_\mu-L_\tau}$ gauge boson for $s_{Z'} \ll 1$, but the corresponding parameter space is excluded by the neutrino trident constraint discussed above for the masses we consider [112].

# 4 Dark matter constraints

If we consider our $Z'$ models to be consistent and realistic, they have to reproduce the observed relic density, or at least predict a sufficiently large annihilation rate after thermal decoupling. We will see that explaining only a fraction of the observed dark matter does not circumvent the constraints, because the typical problem is to reach a large enough dark matter annihilation rate. Given that our model is meant to explain the observed relic density, it then has to respect constraints from indirect and direct detection experiments.

## 4.1 Relic density

Dark matter annihilation is dominated by $Z'$ in the $s$-channel,

$$\chi\bar{\chi} \to Z' \to \text{SM} . \qquad (30)$$

The scalar $S$ has no direct couplings to dark matter, so there is no $S$ mediated annihilation and the scalar only plays a role in the annihilation channel

$$\chi\bar{\chi} \to Z' \to ZS . \qquad (31)$$

For the $U(1)_X$ model, the gauge coupling $g_{Z'}$ and the mixing angle $s_{Z'}$ both enter the annihilation rate, because they determine the $Z'$ coupling to dark matter and SM particles, respectively. In the upper panels of Fig. 7 we show the corresponding parameter space, for which a relic density in the range $(0.3 \dots 1.1) \times 0.12$ [30] is reproduced for $m_{Z'} = 500$ GeV, with $m_\chi = 100$ GeV

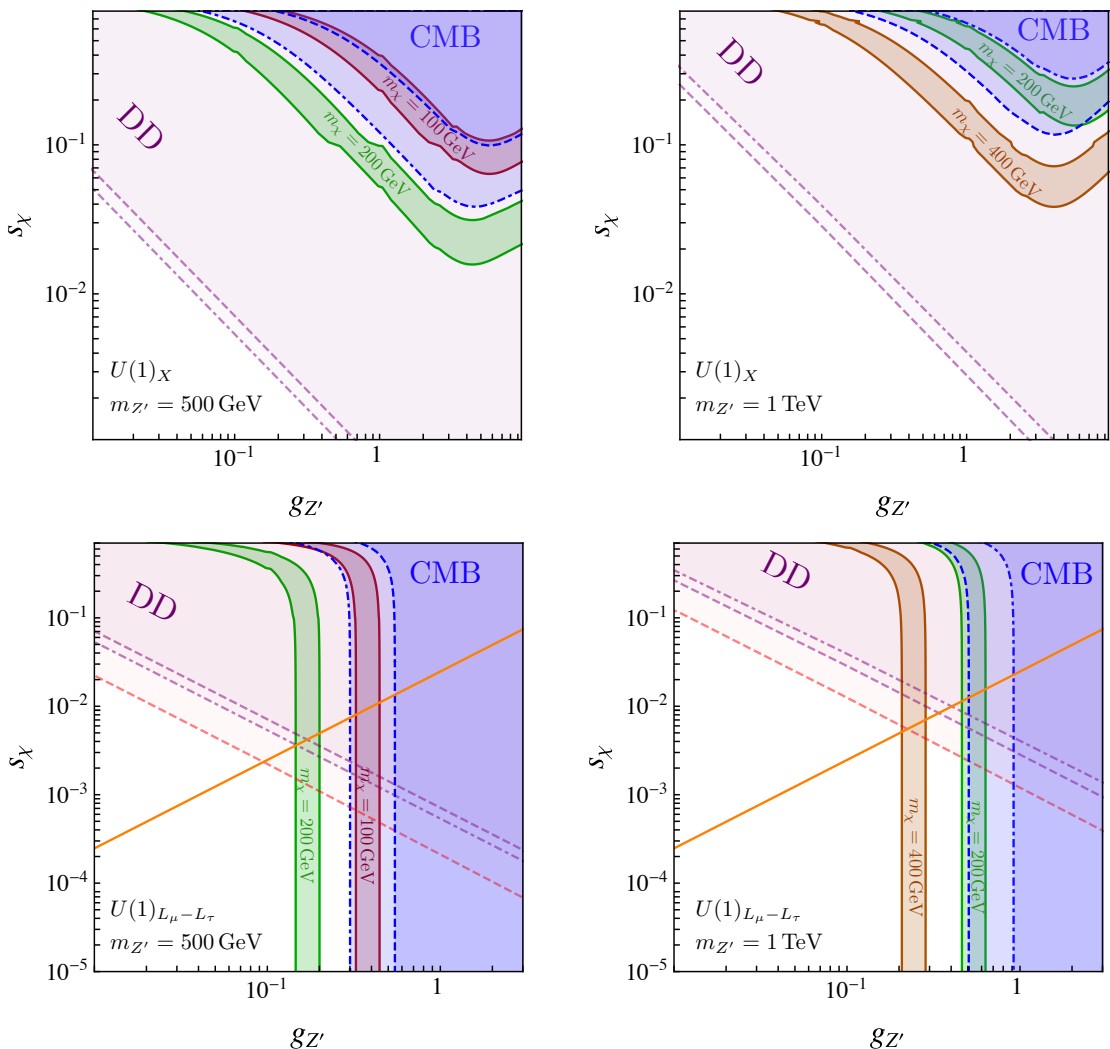

Figure 7: Constraints from the observed relic density in the $U(1)_X$ (upper) and $U(1)_{L_\mu - L_\tau}$ model (lower). We fix $m_{Z'} = 500$ GeV (left) with $m_\chi = 100, 200$ GeV or $m_{Z'} = 1$ TeV (right) with $m_\chi = 200, 400$ GeV. Indirect and direct detection constraints are shaded. The (dot-) dashed contours correspond to $m_\chi = (200)100$ GeV or $m_\chi = (400)200$ GeV, respectively. The Xenon1T projection is indicated by the red dashed contour in the lower panels. The orange line indicates the purely loop-induced mixing angle for a given $g_{Z'}$.

or $m_\chi = 200$ GeV (upper left) and $m_{Z'} = 1$ TeV, with $m_\chi = 200$ GeV or $m_\chi = 400$ GeV (upper right), respectively. Apart from near the $Z'$-pole $m_{Z'} = 2m_\chi$, both couplings need to be sizable to reproduce the observed relic abundance. Generally, a large gauge coupling $g_{Z'}$ allows for smaller mixing angles $s_{Z'}$; only for very large $g_{Z'}$ the mixing angle $s_{Z'}$ has to increase again to introduce a sizable $Z'$ branching ratio to SM particles. Following Eq.(11) this constrains the mass splitting between the $Z'$ and the scalar $S$ mediators. If we assume $m_\chi = 200$ GeV we find that $g_{Z'} = 0.1 \ldots 1$ requires roughly $s_{Z'} = 0.5 \ldots 0.04$, translating into

$$\frac{m_S}{m_{Z'}} = (1 \ldots 6) \sqrt{\lambda_S} \,, \tag{32}$$

if, following Eq.(55), we assume $q_S = 1$ based on the neutrino sector.

In the $U(1)_{L_\mu-L_\tau}$ case, the relic density can be set by $g_{Z'}$ alone, as is evident from the lower panels of Fig. 7. In the absence of additional matter, the mixing angle for a given gauge coupling $g_{Z'}$ is specified by Eq.(16). We indicate this loop-induced value of the mixing angles $s_{Z'}$ for gauge couplings preferred by the relic density by the orange line in the lower panels of Fig. 7. For the mass splitting between the two mediators we now find

$$\frac{m_S}{m_{Z'}} \approx \frac{\sqrt{\lambda_S}}{g_{Z'}} = (0.2 \ldots 0.5)\,\sqrt{\lambda_S}\,. \tag{33}$$

A general bound on the mass of the dark matter candidate arises from the bound on invisible Higgs decays $BR(H \to inv) < 0.23$ [91,92,118,119]. It constrains the loop-induced decay $H \to \chi\bar{\chi}$ through the $H-Z'-Z'$ coupling. We avoid this constraint by assuming $2m_\chi > m_H$.

## 4.2 Indirect detection

If dark matter annihilates into charged leptons, it can be constrained through the cosmic positron flux. The positron spectrum has been measured by HEAT [120], PAMELA [121], FERMI-LAT [122], and AMS [123]. It is most sensitive to dark matter masses around 100 GeV. For heavier dark matter the sensitivity drops rapidly [124,125], and uncertainties in the astrophysical background modeling translate into sizable errors in the production cross section and slope of the measured spectrum [126]. Note that we do not attempt a fit of excesses in PAMELA, FERMI-LAT or AMS [127].

An especially clean test of many dark matter models is provided by measurements of the polarization fluctuation and temperature of the cosmic microwave background (CMB) [128]. Dark matter annihilation during the period of last scattering induces distortions of the CMB spectrum and temperature. Annihilation into charged leptons, in particular electrons, comes with the highest effective deposited power fraction $f_{eff}$. A dominant annihilation channel driven by large kinetic mixing in the $U(1)_X$ and $U(1)_{L_\mu-L_\tau}$ models is

$$\chi\bar{\chi} \to Z' \to e^+e^-\,, \tag{34}$$

driven by the significant coupling of the $Z'$ to the electromagnetic current. For $U(1)_{L_\mu-L_\tau}$ the limit $s_{Z'} = 0$ leaves us with annihilation into muons, taus, and neutrinos. The current limit obtained from Planck data on the annihilation cross section reads [30,129]

$$f_{eff}\,\frac{\sigma v}{m_\chi} \lesssim 3 \times 10^{-28}\,\frac{cm^3}{GeV\,s}\,. \tag{35}$$

A conservative bound assumes 100% annihilation into electrons, unless $s_{Z'} < 0.1$. For $U(1)_{L_\mu-L_\tau}$, we assume a dominant annihilation into muons. The corresponding limits are shown in Fig. 7 shaded blue with dashed and dot-dashed contours for $m_\chi = 100$ GeV and $m_\chi = 200$ GeV ($m_\chi = 200$ GeV and $m_\chi = 400$ GeV), respectively.

## 4.3 Direct detection

Direct detection experiments are sensitive to dark matter scattering off heavy nuclei through $Z'$ exchange, specifically spin-independent scattering in analogy to Higgs exchange. The strongest bounds on spin-independent scattering come from LUX [130], PANDA-X II [131] and Xenon1T [132].

In Fig. 7, we show the constraints obtained by the first Xenon1T results for the $U(1)_X$ extension (upper panels) and the $U(1)_{L_\mu-L_\tau}$ extension (lower panels). The excluded region

is indicated in purple, with dashed and dot-dashed contours for $m_\chi = 100(200)$ GeV and $m_\chi = 200(400)$ GeV, respectively. We further include the projected reach for XenonnT [133, 134] in the lower panels for $m_\chi = 200(400)$ GeV as a dashed red contour.

For both models, the $Z'$ couplings to nuclei are proportional to the kinetic mixing $s_{Z'}$. In the $U(1)_X$ model the values of $s_{Z'}$ necessary to explain the relic density are completely excluded by Xenon1T. In contrast, for the $U(1)_{L_\mu - L_\tau}$ model the relic density can be set by annihilation through the gauge coupling $g_{Z'}$ alone, while the direct detection cross section is proportional to $s_{Z'}$. In absence of a tree-level mixing, the loop-induced mixing given in Eq.(16) is the largest effect from $g_{Z'}$-dependent couplings. Couplings not proportional to the kinetic mixing only arise at the two-loop level [135] and can be neglected. We indicate the value of the loop-induced mixing angles in Fig. 7 as an orange line. For both $m_{Z'} = 500$ GeV and $m_{Z'} = 1$ TeV, a purely loop-induced kinetic mixing allows for an explanation of the observed DM relic density.

# 5 LHC signatures

A key question for $Z'$ mediators at the LHC is how we can establish the link to the dark matter sector once we discover a di-lepton resonance through kinetic mixing. This is complicated by the presence of sizable $Z'$ branching ratios to neutrinos in the $U(1)_X$ and $U(1)_{L_\mu - L_\tau}$ models. We follow two strategies to establish the $Z'$ as a dark matter mediator: a profile analysis of the di-lepton mass peak [138] and a combination with the mono-jet signal. In the case of very small mixing angles the production cross section of the $Z'$ can become smaller than the production cross section of the scalar $S$, whose decays are dominated by the $S \to Z'Z'$ decay rate. We present a third discovery strategy based on the process $S \to Z'Z' \to \mu^+ \mu^- \not{E}_T$.

For any thermal dark matter scenario, the relic abundance strongly constrains the kinetic mixing angle $s_{Z'}$. As discussed in the last section, a $U(1)_X$ gauge boson is excluded as a single mediator through direct detection. For a $U(1)_{L_\mu - L_\tau}$ gauge boson with $m_{Z'} \lesssim 1$ TeV direct detection requires $s_{Z'} \lesssim 0.01$, leading to a suppressed $Z'$ production rate. In addition, the gauge coupling needs to be sizable $g_{Z'} > 0.1$, to allow for an efficient annihilation in the early universe. Following Eq.(11) the scalar $S$ then cannot decouple from the spectrum and will therefore play an important role in the LHC phenomenology.

## 5.1 $Z'$ profile

In both, the $U(1)_X$ and the $U(1)_{L_\mu - L_\tau}$ models, the mediator has a sizable branching ratio into leptons. We can approximately relate the di-lepton production rate to the mono-jet signal via

$$\frac{\sigma(pp \to Z' \to \not{E}_T + \text{jet})}{\sigma(pp \to Z' \to \ell^+ \ell^-)} = \frac{\alpha_s}{4\pi} \frac{\text{BR}(Z' \to \chi \bar{\chi}) + \text{BR}(Z' \to \nu \bar{\nu})}{\text{BR}(Z' \to \ell^+ \ell^-)} . \tag{36}$$

It is safe to assume that any kinetic mixing large enough to observe a mono-jet signal will first give a di-lepton signal.

In this situation, we can use a fit of the $Z'$-width in the di-lepton channel to constrain the $Z'$ branching ratio to dark matter, in analogy to the measurement of the number of light neutrinos at LEP [138]. This measurement heavily relies on the ATLAS and CMS energy resolution for high-energy di-leptons. The lepton energy resolution translates into a resolution of the $Z'$ width at the per-cent level for electrons [137] and several per-cent for muons [139]. In Fig. 8 we compare the experimental resolution for $Z' \to \mu^+ \mu^-$ and $Z' \to e^+ e^-$ to the predicted $Z'$ width in the $U(1)_{L_\mu - L_\tau}$ model (left) and $U(1)_X$ model (right) for $g_{Z'} = 0.25, 0.5, 0.75$. A shape

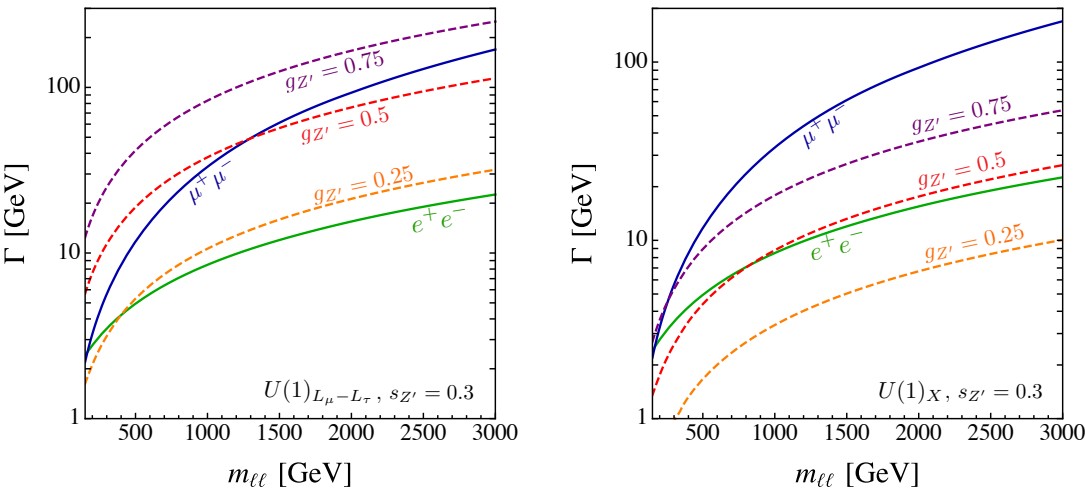

Figure 8: Total $Z'$ width predicted in the $U(1)_X$ and $U(1)_{L_\mu - L_\tau}$ models for $s_{Z'} = 0.3$ and $g_{Z'} = 0.25, 0.5, 0.75$. We also show the detector resolution for $e^+ e^-$ and $\mu^+ \mu^-$ resonances as a function of the di-lepton mass identifying $m_{Z'} = m_{\ell\ell}$.

analysis will only give information on invisible $Z'$ decays if detector resolution is smaller than the total width.

In the $U(1)_{L_\mu - L_\tau}$ model, the branching ratio $\text{BR}(Z' \to e^+ e^-)$ is suppressed by the kinetic mixing $s_{Z'}$. A fit to the $Z'$ width in this channel can still constrain an invisible $Z'$ decay channel to 1% or better.

## 5.2 Invisible $Z'$ decays

An alternative strategy to establish the nature of the $Z'$ as a dark matter mediator is to measure the mono-jet cross section and combine it with the di-lepton rate. The presence of a dark matter coupling strongly enhances the predicted invisible $Z'$ width. For instance, for the $U(1)_X$ model typically $\text{BR}(Z' \to \nu\bar{\nu}) \approx 10\%$ without any coupling to dark matter and $\text{BR}(Z' \to \chi\bar{\chi}) \approx (70\%, 99\%)$ with $\text{BR}(Z' \to \nu\bar{\nu}) \lesssim (3\%, 1\%)$ with a dark matter coupling $s_{Z'} = (0.84, 0.1)$. For a $U(1)_{L_\mu - L_\tau}$ gauge boson, the decay into neutrinos dominates even in the presence of dark matter. Both scale with the gauge coupling $g_{Z'}$, and $\text{BR}(Z' \to \chi\bar{\chi}) \approx (10\%, 20\%)$ for $s_{Z'} = (0.84, 0.1)$. It is therefore necessary to constrain the invisible $Z'$ width to a similar precision to either rule out or establish a link to dark matter.

As usual, invisible mediator decays lead to large missing transverse energy in association with hard jets, Eq.(36). The dominant backgrounds are $Z(\to \nu\nu)$+jets and $W(\to l\nu)$+jets. The latter can be suppressed with a lepton veto, but a fraction of events will remain if the lepton falls outside the detector acceptance or does not meet the isolation requirements. Other channels such as $t\bar{t}$ and $Z(\to ll)$+jets comprise less than 1% of the background and are not considered here.

We simulate the backgrounds with leading-order matrix elements, merged with up to two additional jets in the parton shower using the CKKW-L procedure, as implemented in SHERPA [140]. For the signal we rely on MADGRAPH5 [115] and PYTHIA8 [141]. Both, signal and background samples are passed through the DELPHES [142] detector simulation with the ATLAS default detector card and $R = 0.4$ anti-$k_T$ jets.

As a start, we consider a standard cut-and-count analysis, following an 8 TeV CMS anal-

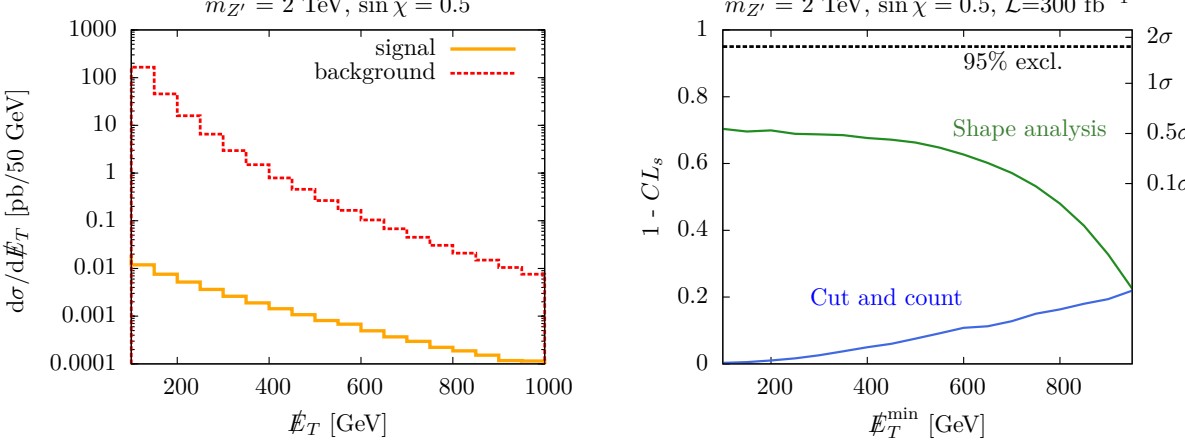

Figure 9: Left: $\not{E}_T$ distribution for a typical signal and the combined $W/Z$+jets background. Right: expected confidence limit and associated Gaussian significance as a function of the $\not{E}_T$ cut using the full $\not{E}_T$ shape information (green) versus a cut-and-count results (blue).

ysis [143]. We require a minimum transverse energy $\not{E}_T > 100$ GeV and a hard jet with $p_T > 100$ GeV and $|\eta| < 2.5$. Events with a second jet only pass if $p_T > 30$ GeV, $|\eta| < 4.5$, and $\Delta\phi(j_1, j_2) < 2.5$, where the last requirement suppresses QCD di-jets. Events with additional jets with $p_T > 30$ GeV and $|\eta| < 4.5$ are vetoed, as are events with one or more isolated leptons.

We select regions with $s/\sqrt{b + (\alpha s)^2 + (\beta b)^2} > 2$, where $\alpha$ and $\beta$ are systematic uncertainties on the signal and background, respectively. The most excluded region is then used to set the limit. The total signal rate is dominated by the low-$\not{E}_T$ regime, with more than 80% of signal events coming from $\not{E}_T < 400$ GeV for our model parameters. This implies that for $300$ fb$^{-1}$ our results are systematics limited, and it is instructive to ask whether the precision can be improved by using the full $\not{E}_T$ shape information of the $\not{E}_T$ distribution.

To this end we perform a binned likelihood analysis of the $\not{E}_T$ distribution. Our procedure is based on the modified frequentist $CL_s$ method [144, 145]. Further details, including the modelling of systematics, can be found in the Appendix. We highlight the improvement over the standard approach in Fig. 9, where we show the expected $CL_s$ limit in a currently allowed parameter point as a function of the minimum $\not{E}_T$ cut, both for a shape analysis and for a cut-and-count analysis. The limit from the shape analysis gradually degrades as more bins are excluded and more information is lost, while the cut-and-count limit moderately improves when we apply a very stringent cut. This shows how a simple counting experiment above a stringent $\not{E}_T$ cut is not the most effective way of observing a mono-jet signal.

The choice of the $\not{E}_T$ distributions can be further optimized by including two-dimensional histograms, provided the proper correlations between variables are available. For example, in Fig. 10 we show the correlation between the first and second jet $p_T$, showing potential discriminating power. In practice, including this information requires full control over the correlations and a very large event sample to obtain a reliable estimate of the event counts, so we merely comment that it is worth pursuing in the future.

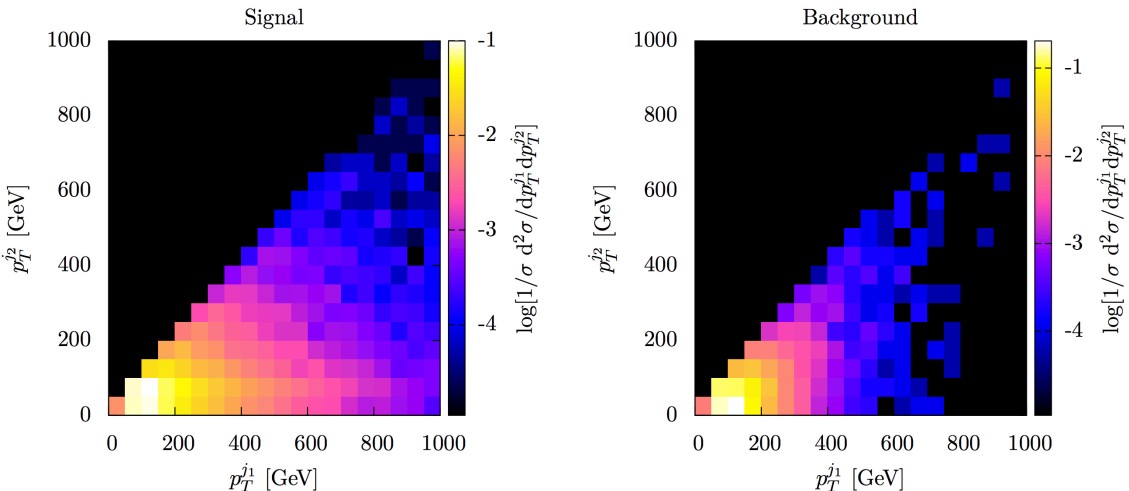

Figure 10: The two-dimensional distributions of the leading and second jet $p_T$ for signal (left; same benchmark point as plotted above) and background (right).

### 5.3 Exploiting $S$ decays

Our consistent model setup allows us to include the scalar mode in the $Z'$ analysis. Given the observed relic density and the direct detection constraints the scalar mass cannot be much larger than the vector mass. Following Sec. 3 the kinetic mixing $s_{Z'}$ is strongly constrained, unless the $Z'$ is very heavy. At least in the $U(1)_{L_\mu - L_\tau}$ case the relic density can be reproduced independently of $s_{Z'}$ through annihilation into leptons. However, a sizable kinetic mixing is necessary to produce the $Z'$, since any coupling between the $Z'$ and protons is proportional to $s_{Z'}$ for both the $U(1)_X$ and $U(1)_{L_\mu - L_\tau}$ models. A simplified model with the $Z'$ mediator and a dark matter candidate does not predict any relevant LHC signal.

In contrast to the kinetic mixing angle, the Higgs portal coupling $\lambda_{HS}$ is not protected for example by an embedding in a non-abelian gauge group. In the absence of an anomaly even without a DM candidate, there is also no reason for the $S$ to couple to the DM. This way the scalar mixing angle is not constrained by direct detection and can be large. This motivated searches for the vector mediator in the process

$$pp \to S \to Z'Z' \,, \tag{37}$$

proportional to the scalar mixing angle $\sin\alpha$ and independent of $s_{Z'}$. Additional searches for $S \to ZZ'$ decays are possible, but the corresponding partial width is again proportional to $s_{Z'}$.

The decay $S \to Z'Z'$ defines a mono-$Z'$ signal [146], allowing for a discovery of a vector mediator through the scalar portal. This signature is established for dark radiation [147] and extended dark sectors [148]. In consistent vector mediator models the mono-$Z'$ signal is resonantly enhanced. Another promising signal is the competing decay

$$S \to Z'Z' \to 4\mu \,. \tag{38}$$

The two signals scale like

$$\frac{\sigma(pp \to S \to \ell^+\ell^- \not{E}_T)}{\sigma(pp \to S \to 4\ell)} \approx \frac{\Gamma(Z' \to \chi\bar{\chi})}{\Gamma(Z' \to \ell^+\ell^-)} \,, \tag{39}$$

with $\Gamma(Z' \to \chi\bar{\chi}) \propto g_{Z'}^2$. On the lepton side, $\Gamma(Z' \to e^+e^-) \propto s_\chi^2$ for both $U(1)_X$ and $U(1)_{L_\mu - L_\tau}$, while $\Gamma(Z' \to \mu^+\mu^-) \propto s_\chi^2$ for $U(1)_X$ and $\Gamma(Z' \to \mu^+\mu^-) \propto g_{Z'}^2$ for $U(1)_{L_\mu - L_\tau}$.

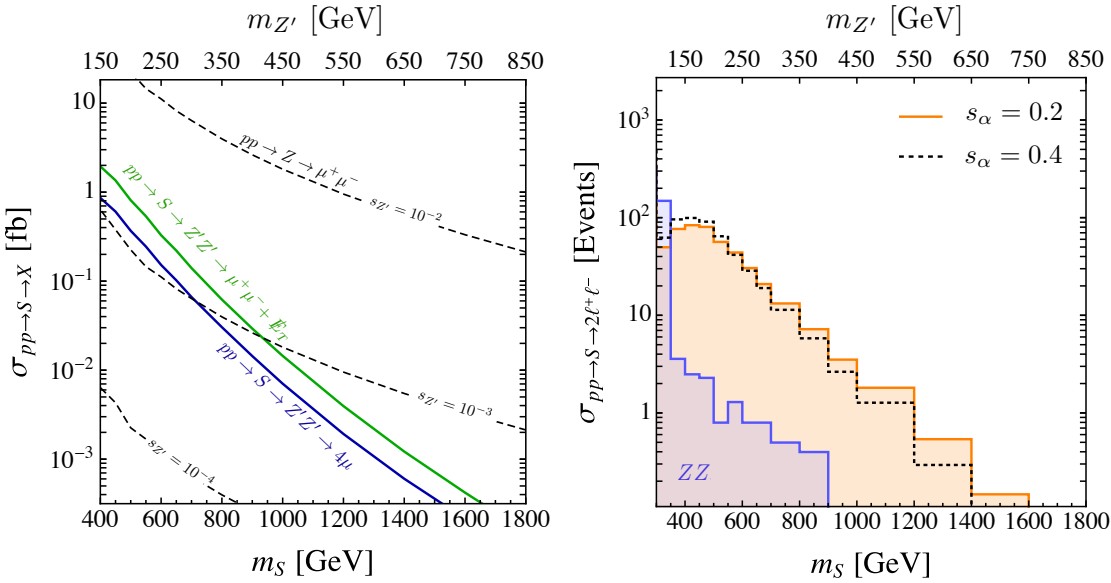

Figure 11: Left: $S$-induced mono-$Z'$ and four-muon signal rates compared to the di-lepton resonance for $s_{Z'} = 10^{-2}, 10^{-3}, 10^{-4}$. Right: signal and background events for $pp \rightarrow S \rightarrow 4\mu$ assuming $s_\alpha = 0.2$ and 0.4 after all cuts.

A measurement of all three decays would allow us to identify the underlying gauge group and constrain the dark matter contribution to the invisible $Z'$ width.

In the left panel of Fig. 11 we see how for $\sin \alpha = 0.4$, $m_S \lesssim 1.8$ TeV, and $s_{Z'} \sim 10^{-3}$, the 4-lepton and mono-$Z'$ cross sections can exceed the di-lepton cross section. We assume a collider energy of 14 TeV. In addition, the signal can be easily extracted through the resonance conditions $m_{\ell\ell} \approx m_{Z'}$ and $m_{Z'Z'} \approx m_S$. In the analysis we ask for two pairs of opposite sign muons reconstructing a $Z'$ each, and implement cuts on the invariant masses

$$m_{4\mu} = (1 \pm 0.1)\, m_S \qquad \text{and} \qquad m_{\mu\mu} = (1 \pm 0.1)\, m_{Z'}\,, \tag{40}$$

as well as $p_{T,\ell} > 20$ GeV for each muon. We show the $S \rightarrow 4\mu$ signal and background rates for an integrated luminosity of 3 ab$^{-1}$, assuming $s_\alpha = 0.2$ and 0.4 in the right panel of Fig. 11. We fix the gauge coupling to the maximum value $g_{Z'} = 0.1 - 0.85$ allowed by the indirect constraints in Sec. 3. The blue contours show the dominant $ZZ$ backgrounds after cuts. In the lowest mass bin, the overlap with the $Z$ resonance is responsible for the spike in background events. Smaller scalar mixing angles do not necessarily result in fewer signal events once we take into account the scaling of the decay widths $\Gamma(S \rightarrow \text{SM}) \propto s_\alpha^2$ and $\Gamma(S \rightarrow Z'Z') \propto g_{Z'}^2$. An increased production rate is partially cancelled by a reduced branching ratio BR($S \rightarrow Z'Z'$).

The mono-$Z'$ signal rate is larger than the 4-lepton rate by an order of magnitude throughout the parameter space. The $p_{T,\ell\ell}$ spectrum of the signal displays a Jacobian peak characteristic for the resonant decay. The maximal value

$$p_{T,\ell\ell}^{\max} \approx m_S \left( \frac{1}{4} - \frac{m_{Z'}^2}{m_S^2} \right)^{1/2}, \tag{41}$$

allows us to reduce the backgrounds through harder $\slashed{E}_T$ cuts. We show the $p_{T,\ell\ell}$ distribution for $(m_S = 500, m_\chi = 200)$ GeV and $(m_S = 350, m_\chi = 150)$ GeV. We apply the cuts from Ref. [149]

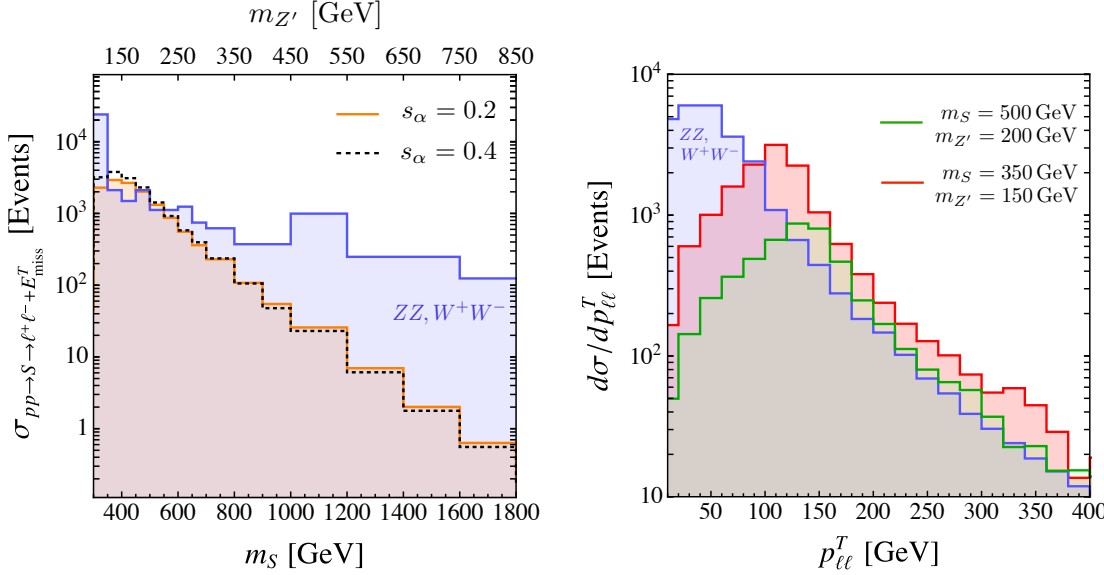

Figure 12: Left: signal and background events for $pp \to S \to \mu^+\mu^- + \not{E}_T$ assuming $s_\alpha = 0.2$ and 0.4, after all cuts. Right: signal and background rates for two benchmark points.

and in addition require

$$\not{E}_T > 100\,\text{GeV} \qquad \text{and} \qquad p_{T,\ell\ell} > \begin{cases} 60\,\text{GeV} & m_S < 600\,\text{GeV} \\ 100\,\text{GeV} & m_S > 600\,\text{GeV} \end{cases}. \qquad (42)$$

The hardest lepton pair has to reconstruct the $Z'$ mass to ±10%. The signal and background are shown in the left panel of Fig. 12 for $s_\alpha = 0.2$ and 0.4 for different masses $m_S$ and $m_{Z'}$ and gauge couplings $g_{Z'} = 0.1 - 0.85$. Again, the overlap with the $Z$ resonance leads to the large number of background events in the first bin. Even for a soft $\not{E}_T$ cut the signal will be even more significant than the 4-lepton signal because of the large signal rate.

In Fig. 13, we show the significances of the two $S$-induced signals for $s_\alpha = 0.2$ and 0.4. For the small kinetic mixing angles implied by indirect constraints and direct detection, the mono-$Z'$ signal can be the discovery channel for a $U(1)_{L_\mu - L_\tau}$ mediator. Note that the results of this section also hold for the gauge groups $U(1)_{L_e - L_\tau}$ and $U(1)_{L_e - L_\mu}$ for $s_{Z'} \to 0$, taking into account the LEP bounds of Eq.(21).

It is clear from Fig. 13 that a simple cut-and-count analysis offers little sensitivity above $m_s \simeq 1$ TeV, even after applying cuts for an on-shell $Z'$. Therefore, analogous to Sec. 6.2 we apply a shape analysis of the $p_{T,\ell\ell}$ spectrum shown in Fig. 12[1]. We see a moderate gain from the shape analysis, since the distinctive Jacobian peak of the signal offsets the drop in sensitivity from the reduction in cross section, however the improvement is less substantial than in the mono-jet case, since the resonance cuts already suppress the background quite effectively.

# 6 Conclusions

The best-motivated simplified models for dark matter with a vector mediator are anomaly-free, gauged global symmetries of the SM. We discuss several different such gauge groups, a $U(1)_X$

---

[1]We do not perform the shape fit below $m_S < 900$ GeV where the cut-and-count significance is already high enough to test the presence of a signal.

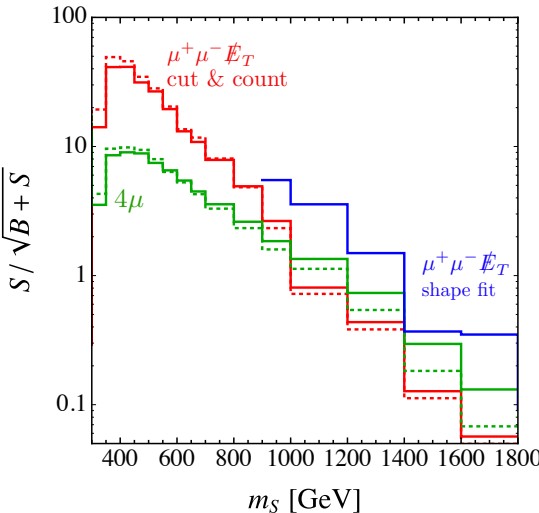

Figure 13: Significance of the mono-$Z'$ signal in red in comparison to the 4-muon signal in green for difference scalar masses and for $s_\alpha = 0.2(0.4)$ shown by the solid (dotted) contours, respectively. The red (blue) contours correspond to the significance based on the cut & count (shape fit) analysis.

under which only dark matter is charged and all couplings to the SM are mediated through a kinetic mixing term, charged lepton family number differences $U(1)_{L_e-L_\mu}$, $U(1)_{L_e-L_\tau}$ and $U(1)_{L_\mu-L_\tau}$, and the gauged baryon-lepton number difference $U(1)_{B-L}$. Obviously, mediators with tree-level couplings to electrons are strongly disfavored by LEP bounds, leaving us with $U(1)_X$ and $U(1)_{L_\mu-L_\tau}$ for a detailed study.

For the $U(1)_X$ model sizable kinetic mixing angles are necessary to reproduce the observed relic density, which brings the model into conflict with direct detection bounds. For the $U(1)_{L_\mu-L_\tau}$ mediator the relic density can be explained for sub-TeV masses and order-one gauge couplings. Even allowing for loop-induced kinetic mixing this parameter space is compatible with constraints from Planck measurements of the CMB spectrum and direct detection. However, the dark matter phenomenology constrains the mass splitting between the vector mediator and the scalar mediator responsible for the $Z'$ mass generation.

A common feature of the gauge groups we consider is a sizable branching ratio BR($Z' \to \nu\bar\nu$). This introduces a mono-jet signal even in the absence of a dark matter coupling. We discuss the prospects of observing decays to dark matter by fitting the $Z'$-width in the di-lepton channel and by precisely measuring the mono-jet rate. In principle, the former is much more sensitive. However, for $m_{Z'} \approx 1$ TeV the ATLAS and CMS energy resolution rule out this method for $\Gamma_{Z'} < 5(100)$ GeV for electrons (muons). In this case a precise measurement of the mono-jet rate is indispensable to establish mediator nature of the $Z'$ gauge boson. We explore the additional sensitivity gained by a shape analysis of the $\not{E}_T$ distribution compared to a cut-and-count analysis.

For the small kinetic mixing angles preferred by the dark matter constraints, the $s$-channel production of the $Z'$ mediator at the LHC is strongly suppressed. In contrast, the scalar mediator mode can be produced through a Higgs portal. Since $U(1)_{L_\mu-L_\tau}$ is anomaly free within the SM, the scalar does not have to couple to the dark matter. Its dominant decay is $S \to Z'Z'$, if kinematically allowed. The corresponding signatures are a resonantly enhanced 4-lepton signal $pp \to S \to Z'Z' \to 4\mu$ and a mono-$Z'$ signal $pp \to S \to Z'Z' \to \mu^+\mu^-\not{E}_T$. This combination is characteristic for a consistent vector mediator model based on this gauge group. In particular the mono-$Z'$ final state with a leptonic $Z'$ decay is a potential discovery channel for

our consistent vector mediator model.

## Acknowledgments

We thank Julian Heeck for useful comments regarding the structure of the neutrino mass matrices and Dirk Zerwas for reminding us of the correct lepton energy resolutions.

## A  Details of $U(1)$ extensions

The scalars in (9) acquire VEVs $\langle H \rangle = v/\sqrt{2}$ and $\langle S \rangle = v_S/\sqrt{2}$, and the Higgs portal term induces the mixing

$$\mathcal{M}^2_{H,S} = \begin{pmatrix} \lambda_H v^2 & \lambda_{HS} v v_S \\ \lambda_{HS} v v_S & \lambda_S v_S^2 \end{pmatrix} . \tag{43}$$

It can be diagonalized with a unitary rotation

$$\begin{pmatrix} S \\ H \end{pmatrix} \rightarrow \begin{pmatrix} c_\alpha & s_\alpha \\ -s_\alpha & c_\alpha \end{pmatrix} \begin{pmatrix} S \\ H \end{pmatrix} \qquad \text{with} \qquad t_{2\alpha} = \frac{2\lambda_{HS} v v_S}{\lambda_H v^2 - \lambda_S v_S^2} , \tag{44}$$

where $t_{2\alpha} \equiv \tan(2\alpha)$.

The interaction with the SM-gauge sector allows for a mixed kinetic term involving the Standard Model $U(1)_Y$-boson as given in (6), where the notation $\hat{B}_{\mu\nu}$ indicates that the kinetic terms of the gauge fields are not yet canonically normalized. As indicated by the above notation with $s_{Z'} \equiv \sin\theta_{Z'}$ we consider kinetic mixing a phenomenon related to field rotations, but the term $s_{Z'}$ in the Lagrangian does not arise from a rotation. Instead, it is generally allowed by all symmetries at tree level and will typically appear at one loop, even if it should vanish at tree level. We assume $s_{Z'} < 1$, otherwise the Lagrangian in Eq.(6) corresponds to a theory with a single propagating gauge boson ($s_{Z'} = 1$) or a kinetic term with the wrong sign ($s_{Z'} > 1$).

For the abelian case the kinetic term can be diagonalized by an orthogonal rotation in the two gauge fields. The problem with such an orthogonal transformation is that it shifts the hypercharge and eventually the electromagnetic current. To explicitly keep the electromagnetic current and the canonical normalization, we introduce a non-orthogonal rotation $G(\theta_{Z'})$ instead,

$$\begin{pmatrix} \hat{B}_\mu \\ \hat{Z}'_\mu \end{pmatrix} = G(\theta_{Z'}) \begin{pmatrix} B_\mu \\ Z'_\mu \end{pmatrix} = \begin{pmatrix} 1 & -s_{Z'}/c_{Z'} \\ 0 & 1/c_{Z'} \end{pmatrix} \begin{pmatrix} B_\mu \\ Z'_\mu \end{pmatrix} . \tag{45}$$

Now the SM fermions couple to the new gauge boson with a coupling strength

$$j'_\mu \rightarrow \frac{1}{c_{Z'}} j'_\mu - t_{Z'} j^Y_\mu , \tag{46}$$

where $j^Y_\mu$ denotes the hypercharge current. The combined mass matrix for the three electroweak gauge bosons $B_\mu$, $W^3_\mu$, and $Z'_\mu$ reads

$$\mathcal{M}^2_{B,W,Z'} = \frac{v^2}{4} \begin{pmatrix} g'^2 & -g g' & -g'^2 t_{Z'} \\ -g g' & g^2 & g g' t_{Z'} \\ -g'^2 t_{Z'} & g g' t_{Z'} & 2g^2_{Z'} \frac{q^2_S v^2_S}{v^2 c^2_{Z'}} + g'^2 t^2_{Z'} \end{pmatrix} , \tag{47}$$

where $g$ and $g'$ denote the $SU(2)_L$ and $U(1)_Y$ gauge couplings. This mass matrix can be diagonalized through a combination of two block-diagonal rotations with the weak mixing angle $\theta_w$ and an additional angle $\theta_3$ in the lower-right block. The mixing angle $\theta_3$ is then given by

$$\tan(2\theta_3) = \frac{2s_{Z'}c_{Z'}s_w v^2(g^2 + g'^2)}{c_{Z'}^2 v^2(g^2 + g'^2)(1 - s_w^2 t_{Z'}^2) - 2g_{Z'}^2 q_S^2 v_S^2}$$
$$= -\frac{2s_{Z'}c_{Z'}s_w}{2g_{Z'}^2 q_S^2} \frac{v^2}{v_S^2} \left(g^2 + g'^2\right) + \mathcal{O}\left(\frac{v^4}{v_S^4}\right). \tag{48}$$

The physical gauge boson masses

$$m_\gamma = 0$$
$$m_{Z,Z'}^2 = \frac{1}{8c_{Z'}^2}\Bigg[ c_{Z'}^2 v^2(g^2 + g'^2) + g'^2 s_{Z'}^2 v^2 + 2g_{Z'}^2 q_S^2 v_S^2$$
$$\pm \sqrt{\left(c_{Z'}^2 v^2(g^2 + g'^2) + g'^2 s_{Z'}^2 v^2 + 2g_{Z'}^2 q_S^2 v_S^2\right)^2 + 8c_{Z'}^2 g_{Z'}^2 q_S^2 v^2 v_S^2(g^2 + g'^2)}\Bigg]$$
$$= \begin{cases} \dfrac{v^2}{4}(g^2 + g'^2)\left(1 - \dfrac{v^2}{v_S^2}\dfrac{s_{Z'}^2 g'^2}{8g_{Z'}^2 q_S^2}\right) + \mathcal{O}\left(\dfrac{v^6}{v_S^4}\right) \\[3mm] \dfrac{g_{Z'}^2 q_S^2 v_S^2}{2c_{Z'}^2} + \dfrac{v^2}{4}g'^2 t_{Z'}^2 + \mathcal{O}\left(\dfrac{v^4}{v_S^2}\right). \end{cases} \tag{49}$$

We show approximate results for $v_S > v$, motivated by our expectation $m_{Z'}, m_S > m_Z$. The alternative series in terms of a small mixing angle $s_{Z'}$ would have to be motivated by specific model considerations [89].

A combination of all three rotations by the kinetic mixing parameter and the angles $\theta_w$, $\theta_{Z'}$, and $\theta_3$ appears in the couplings of the fermionic currents to the boson mass eigenstates,

$$\left(e j_{\text{em}}, \frac{e j_Z}{s_w c_w}, g_{Z'} j_{Z'}\right)\begin{pmatrix} \hat{A} \\ \hat{Z} \\ \hat{Z}' \end{pmatrix} = \left(e j_{\text{em}}, \frac{e}{s_w c_w} j_Z, g_{Z'} j_{Z'}\right) K \begin{pmatrix} A \\ Z \\ Z' \end{pmatrix}$$
$$K = \left[R_1(\theta_3)R_2(\theta_w)G^{-1}(\theta_{Z'})R_2(\theta_w)^{-1}\right]^{-1}$$
$$= \begin{pmatrix} 1 & -c_w s_3\, t_{Z'} & -c_w c_3\, t_{Z'} \\ 0 & c_3 + s_w s_3 t_{Z'} & c_3 s_w t_{Z'} - s_3 \\ 0 & s_3/c_{Z'} & c_3/c_{Z'} \end{pmatrix}. \tag{50}$$

The interesting aspect is that the combination of all angles is not an orthogonal rotation. This is why the electromagnetic fermion current of SM fermions couples to all three gauge bosons.

Similarly, the complex mixing pattern affects the otherwise simple coupling structure of the gauge boson to the two scalars

$$\begin{pmatrix} A & Z & Z' \end{pmatrix}\begin{pmatrix} 0 & 0 & 0 \\ 0 & & \\ 0 & & W \end{pmatrix}\begin{pmatrix} A \\ Z \\ Z' \end{pmatrix}, \tag{51}$$

with the massive sub-matrix

$$W = -\frac{vs_\alpha}{8}\begin{pmatrix} (g^2 + g'^2) & (g^2 + g'^2)s_w t_{Z'} \\ (g^2 + g'^2)s_w t_{Z'} & (g^2 + g'^2)t_{Z'}^2 s_w^2 - \frac{4g_{Z'}^2 q_S^2}{t_\alpha c_{Z'}^2}\frac{v_S}{v} \end{pmatrix} S$$

$$+ \frac{vc_\alpha}{8}\begin{pmatrix} (g^2 + g'^2) & (g^2 + g'^2)s_w t_{Z'} \\ (g^2 + g'^2)s_w t_{Z'} & (g^2 + g'^2)t_{Z'}^2 s_w^2 + \frac{4g_{Z'}^2 q_S^2 t_\alpha}{c_{Z'}^2}\frac{v_S}{v} \end{pmatrix} H + \mathcal{O}\left(\frac{v^2}{v_S}\right). \tag{52}$$

This matrix induces new couplings between the scalars $H$ or $S$ and the gauge bosons $Z$ and $Z'$. They follow a generic hierarchy of couplings

$$\frac{g_{SZZ'}}{g_{HZZ'}} \propto t_\alpha \approx \frac{1}{3}, \tag{53}$$

because the scalar mixing angle is constrained by Higgs coupling strength measurements $\sin\alpha < 0.3$ [91, 92].

It is instructive to link those three gauge groups to neutrino masses [69–75]. For gauged $U(1)_{L_i - L_j}$ symmetries the three lepton generation carry different charges, which implies that the leptons cannot mix and the Yukawa matrix is diagonal. The same is true for the neutrinos, once we add right-handed neutrinos only charged under the new gauge group. The right-handed neutrinos also have a Majorana mass. For example in the case of $U(1)_{L_\mu - L_\tau}$ such a Majorana mass term can appear as the $(e, e)$ entry and in the $(\mu, \tau)$ and $(\tau, \mu)$ entries. In addition, terms of the kind $yNNS$ lead to Majorana masses when the new scalar is replaces by its VEV. Still, $S$ is charged under the new $U(1)$ group, which leads to possible $(e, \mu)$ and $(e, \tau)$ entries. The corresponding, symmetric Majorana mass matrix for three generations of neutrinos reads

$$\begin{pmatrix} m_e & y_{e,\mu}v_S & y_{e,\tau}v_S \\ y_{e,\mu}v_S & 0 & m_{\mu,\tau} \\ y_{e,\tau}v_S & m_{\mu,\tau} & 0 \end{pmatrix}, \tag{54}$$

assuming

$$q_S = 1. \tag{55}$$

As a consequence of the diagonal mass matrices for the charged leptons, the $Z'$ gauge boson has no lepton-flavor violating couplings to charged leptons and flavor-changing neutral currents only arise at the one-loop level. From this construction it is clear that the generation-universal groups $U(1)_X$ and $U(1)_{B-L}$ do not have this direct link to neutrino masses.

## B  Mono-jet shape analysis

A shape analysis like the one discussed in Sec. 5.2 typically distinguishes a background-only hypotheses $H_0$ from a signal-plus-background hypothesis $H_1$. The Neyman-Pearson lemma states that the most powerful test statistic is the likelihood ratio. For a counting experiment in the absence of systematic uncertainties it is given by Poisson probabilities for obtaining $d$ data events given the expectation values $s + b$ and $b$. In practice, we usually take its logarithm,

$$-2\log Q = -2\log\frac{P(d|s+b)}{P(d|b)} = -s + d\log\frac{s+b}{b}. \tag{56}$$

In this form we can easily combine different channels of bins of a distribution and therefore perform a shape analysis for example of a $\not{E}_T$ distribution.

To compute confidence levels we numerically evaluate the corresponding $p$-values by generating a large number of Monte Carlo pseudo-experiments, with $CL_{s+b}$ being the fraction of pseudo-experiments that generate at least as many events as observed in the data. Instead of excluding regions for which $CL_{s+b} \leq 0.05$, we take the $CL_s$ procedure [144, 145], which only excludes this hypothesis if $CL_{s+b}/(1 - CL_b) \leq 0.05$. This is more robust against spuriously high sensitivity when both $s$ and $b$ are small, at the price of being conservative otherwise.

One way of including systematic uncertainties is by convoluting the individual Poisson likelihoods in Eq. 56 with Gaussians. This procedure reduces the sensitivity by smearing the log-likelihood distributions for the two hypotheses, thus reducing the distinction between $s$ and $s + b$.

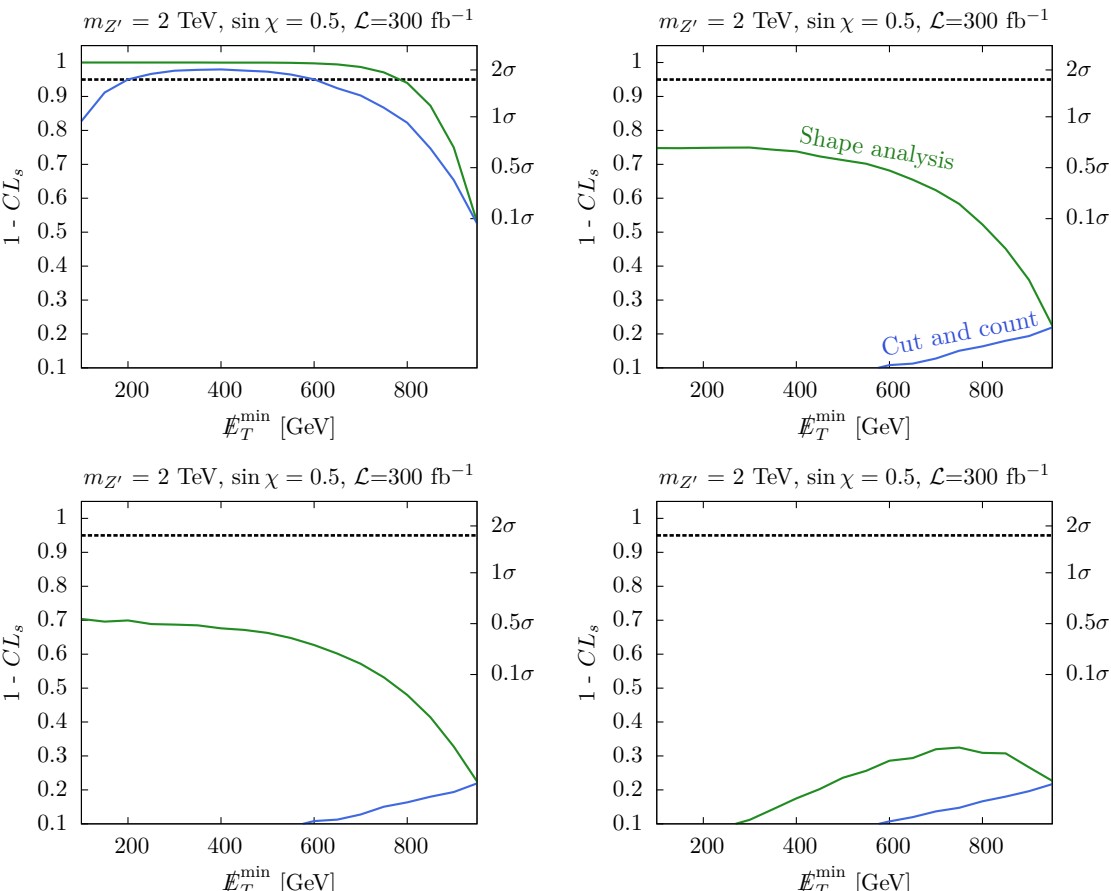

Figure 14: Expected $CL_s$ for excluding the signal hypothesis from a $\not{E}_T$ distribution in mono-jet events as a function of minimum $\not{E}_T$. We show the results from a full shape analysis (green) vs counting all events above the cut as a single bin (blue). Four systematics scenarios are considered: no systematics (top-left), an uncorrelated 5% per-bin background uncertainty (top-right), a 5% per-bin background uncertainty plus 100% correlation between neighbouring bins (bottom-left), and a 5% uncertainty fully correlated across all bins (bottom-right).

Clearly, the separation between the hypotheses and thus the final confidence level is extremely sensitive to the modelling of systematic uncertainties. Therefore it is crucial to cor-

rectly propagate systematics in the limit-setting procedure when using the full shape information from binned distributions. We study four scenarios, in order of increasing conservatism: (i) no systematics at all; (ii) uncorrelated bin-by-bin systematics; (iii) a 5% correlation between each bin and its nearest neighbor with all other correlations zero; and (iv) a flat systematic fully correlated across all bins.

As input data we use the binned mono-jet $\displaystyle{\not}E_T$ distributions for the signal and the combined $Z$+jets and $W$+jets background for 300 fb$^{-1}$ of data. As benchmark point for the test hypothesis, we consider the $U(1)_X$ model discussed in Sec. 5.2 for a $Z'$ mass of 2 TeV and mixing angle $\sin \chi = 0.5$. In Fig. 14 we show $CL_s$ as a function of a minimum $\displaystyle{\not}E_T$ cut for each of the four systematics scenarios, both using the full shape information and using the integrated rate only (cut and count).

Beginning with the unrealistic case of no systematics we see that the full shape analysis provides much more sensitivity than the cut-and-count analysis in the low $\displaystyle{\not}E_T$ region, reflecting the much larger background there. For an uncorrelated 5% systematic on the background in each bin we see a lower significance for both shape and rate analyses, but using shape information carries much better discriminating power than cutting on $\displaystyle{\not}E_T$ and counting events.

To estimate the effects of bin migration, we then include a full correlation between neighbouring bins, with all other correlation coefficients set to zero. This has a mild influence on the significance from the shape analysis, but does not affect our conclusion that the full shape information is a more powerful discriminator. Finally, we consider the extreme scenario of full correlations across all bins. Adding more bins below $\sim 700$ GeV now leads to less discriminating power, because the 5% uncertainty on the background in the low-$\displaystyle{\not}E_T$ region is smeared across all bins. The behavior turns over around $\displaystyle{\not}E_T = 700$ GeV, where statistics becomes the main driver of discriminating power.

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
