# Peer review of "Dark Matter in Anomaly-Free Gauge Extensions"

_SciPost Physics, doi:SciPost Phys. 5, 036 (2018)_

## Round 2 · Referee Report · Anonymous (Referee 1) · 2018-6-28

Strengths

Overall, the paper is very timely and of decent quality. Various formulae are explicitly derived and discussed. Also the level of the numerical studies performed is adequate. Thus in principle I support the publication in SciPost. As far as I can see the considered limits are indeed the relevant ones and the presentation of possible signatures is adequate. In particular the quantitative estimate obtained by a shape analysis instead of a cut-and-count analysis in the monojet phase-space is interesting.

Weaknesses

There are a few minor weaknesses in the presentation of the results and I invite the authors to address these issues.

1-The information density of the paper at hand is quite high (which of course is not necessarily a weaknesses, see above), however the readability of the paper would strongly profit moving some of the detailed derivations to appendices. I invite the authors to implement such changes, as detailed below.

2- The Introduction is partly hard to comprehend. For example I do not understand the relevance of the abrupt sentence "Even if we model our dark matter candidate after the supersymmetric neutralino..."

3- The paper introduces three U(1) models, however very early on in the discussion of relevant limits, one of them turns out to be basically excluded (the B-L model) and the authors state: "This is why at this stage we will drop the U(1)B−L gauge group (and any other group with gauged electrons) from our analysis.". Still, later on again constraints for the B-L are discussed.

4- I do not understand how the program MATRIX is used in the analysis at hand. The authors state: "We compute the Z′ production cross section with MadGraph5 [43], accounting for higher order corrections using Matrix ". To the best of my knowledge the MATRIX Monte Carlo does not provide predictions in any Z' model.

5- There are several typos in the manuscript, starting from the very first sentence: "The nature of dark matter is one the great mysteries", amongst others also including "This setup is by trivially free of anomalies" on page 4 and "the the" on page 15, "gauge bosons the LHC production cross section" on page 17.

Report

The paper introduces three models for particle dark matter based on a spin-1/2 dark matter candidate and a spin-1 mediator. The authors argue that such models should be anomaly free and should also introduce an additional scalar from gauge boson mass generation. In this war the authors suggest that several simplified models used as benchmarks for ongoing Dark Matter searches are inconsistent and incomplete. The authors discuss various theoretical and experimental limits and show that large parts of the spin-1 parameter space is excluded once consistent models are introduced. In particular they consider three signatures that can arise at the LHC.

Requested changes

Following the listed weaknesses above I suggest to

1- Streamline Indoduction

2- Move detailed derivations to appendices, in particular from Section 2.

3- Sections 2 & 3 should be merged, they both discuss the phenomenology and motivation for the three models.

4- Already in this new Section 2 (see above) it should be mentioned that the B-L is basically excluded. In fact I don't necessarily see the point this model is discussed in great detail including several Figures.

5- The use of the code MATRIX should be clarified.

6- At the beginning of Section 6 the authors state: "We follow two strategies to establish the Z′ as a dark matter mediator: a profile analysis of the di-lepton mass peak [63] and a combination with the mono-jet signal." In fact there is a third signatures being discussed in Section 6.3.

7- The authors should state clearly that a realistic uncertainty estimate of the proposed shape analysis is highly non-trivially due to the nature how the monojet backgrounds are constrained.

8- Fix typos

---

## Round 2 · Referee Report · Anonymous (Referee 2) · 2018-7-5

Strengths

  1. The main topic of the paper (dark matter in gauge extended models) is interesting.

  2. The discussion of the mono-jet shape analysis (and its comparison with the cut and count analysis) and of the possibility to establish the link between a $Z^\prime$ and dark matter (even in the presence of $Z^\prime$ decays to invisible neutrinos) is novel and interesting.

Weaknesses

  1. The discussion of the several bounds (Secs. 4, 5) is not particularly novel and it mainly summarizes many results already reported in the literature.

  2. The presentation of the $U(1)_{B-L}$ model is rather misleading: the authors drop this model after an initial short discussion.

Report

The paper discusses dark matter models based on new anomaly-free symmetries which include a fermionic dark matter, a new $U(1)$ gauge boson and a scalar giving mass to the corresponding $Z^\prime$. The authors analyze the several bounds coming from dark matter experiments, as well as colliders and low energy experiments. Furthermore, they compare LHC shape and cut and count analyses to probe the nature of the $Z^\prime$.

Requested changes

  1. The presentation would improve if the authors shorten a bit the discussion of the models in Secs. 2,3 and of the bounds in Secs. 4,5 since these sections are not particularly novel.

  2. At the end of Sec.4, the authors should change the discussion of the LHC $Z\to4\mu$ bound. As shown in their Ref. 34 (1406.2332) and Ref. 19 (1511.04107), as well as in the corresponding CMS analysis http://inspirehep.net/record/1676064/files/EXO-18-008-pas.pdf the $Z\to4\mu$ search can be more powerful than the trident bound in certain regions of parameter space.

---

## Round 3 · Author Response

We thank the referees and the editor for the useful and constructive remarks and criticism on our paper. Together with this report, we submit a revised version of our manuscript taking into account the points raised by all referees. We address them in order. In doing so, we summarize comments on the same subject from the editor and both referees. (Referee/Editor suggestions are in double quotes).
"Please tighten up the presentation of model in Secs 2 and 3, in particular shorten considerably the discussion of the U(1)_{B-L} model and only refer to it. You may want to shift some of the derivations verbatim to appendices, as one of the referees suggested. "
We considerably shortened the discussion in Section 2 and further follow the recommendation of referee 2
"1- Streamline Introduction
2- Move detailed derivations to appendices, in particular from Section 2.
3- Sections 2 & 3 should be merged, they both discuss the phenomenology and motivation for the three models."
and streamline the introduction, move all non-essential steps in the new Appendix A and merge Sections 2&3.
We consider the discussion of the U(1)B-L model in the new Section 2 as essential to explain the motivation for discarding this model later. We understand the criticism that this might suggest an equal treatment of the three models during the rest of the paper and we therefore put a clarifying statement already in the introduction:
'We argue that searches for missing energy signals at the LHC are particularly powerful for two of these models, namely the $U(1)X$ and the $U(1)$ gauge groups. After deriving the properties of the mediators for the three classes of models defined above, we focus our analysis on these two models. '
"2 Also shorten the discussion of bounds in Secs. 4,5 by summarizing the findings of the previous studies quoted there. Please take into account the comment of one of the two referees ("As shown in their Ref. 34 (1406.2332) and Ref. 19 (1511.04107), as well as in the corresponding CMS analysis http://inspirehep.net/record/1676064/files/EXO-18-008-pas.pdf the Z→4μ search can be more powerful than the trident bound in certain regions of parameter space."). "
Even though many of the constraints discussed in Section 4 and 5 are not new, we consider it important to explicitly state what constraints we include in our analysis (and in the plots). To the best of our knowledge a comparative study of all the different models discussed in this section has not been presented elsewhere. We therefore only streamline these sections in a minor way in the revised submission.
Referee I is right in pointing out that the Z -> 4mu search at CMS provides stronger constraints for the L_mu-L_tau gauge boson for masses m_Z’ < 60 GeV. This is outside the interesting parameter space for our analysis and the region shown in Fig.5 and we state so explicitly in the revised version.
"3. Please respond to the referee comment concerning the number of signatures for the Z' as DM mediator."
The Referee is right that this sentence in the introduction of Section 5 omitted the last subsection of this part of the paper. In the revised version we explicitly refer to the third part of this section:
'In the case of very small mixing angles the production cross section of the $Z'$ can become smaller than the production cross section of the scalar $S$, whose decays are dominated by the $S\to Z'Z'$ decay rate. We present a third discovery strategy based on the process $S\to Z'Z'\to \mu^+\mu^- \met$.'
"4. Please detail how you modified the matrix code."
'The matrix code is used to derive the NNLO production cross section for the Z boson at the LHC and we use the corresponding K factor to account for corrections to the Z’ production cross section. Since the production cross section for the U(1)X and the U(1) gauge boson are only produced through kinetic mixing with the Z at the LHC, this estimate should be good. We added an explanatory sentence in the revised version.'
"5. Please make statements concerning uncertainty estimates more explicit, as one of the referees suggested.
The authors should state clearly that a realistic uncertainty estimate of the proposed shape analysis is highly non-trivially due to the nature how the monojet backgrounds are constrained."
In the appendix discussing the details of the shape analysis, we explicitly state
'Clearly, the separation between the hypotheses and thus the final confidence level is extremely sensitive to the modeling of systematic uncertainties.'
"6. Please check, again, for typos."
We thank the referees for pointing out the typos, and hope we have eliminated all of them.
We hope that after addressing all points raised by the referees, our paper will be considered fit for publication by SciPost.

---

## Round 3 · List of Changes

see comments.

You are currently on this page

---

## Editorial Decision

published